# Duplex DNA and BLM regulate gate opening by the human TopoIIIα-RMI1-RMI2 complex

Julia A. M. Bakx[1,4], Andreas S. Biebricher[1,4], Graeme A. King[1,3,4], Panagiotis Christodoulis[1], Kata Sarlós[2],
Anna H. Bizard [2], Ian D. Hickson [2], Gijs J. L. Wuite [1,5✉] & Erwin J. G. Peterman [1,5✉]

Topoisomerase IIIα is a type 1A topoisomerase that forms a complex with RMI1 and RMI2 called TRR in human cells. TRR plays an essential role in resolving DNA replication and recombination intermediates, often alongside the helicase BLM. While the TRR catalytic cycle is known to involve a protein-mediated single-stranded (ss)DNA gate, the detailed mechanism is not fully understood. Here, we probe the catalytic steps of TRR using optical tweezers and fluorescence microscopy. We demonstrate that TRR forms an open gate in ssDNA of 8.5 ± 3.8 nm, and directly visualize binding of a second ssDNA or double-stranded (ds)DNA molecule to the open TRR-ssDNA gate, followed by catenation in each case. Strikingly, dsDNA binding increases the gate size (by ~16%), while BLM alters the mechanical flexibility of the gate. These findings reveal an unexpected plasticity of the TRR-ssDNA gate size and suggest that TRR-mediated transfer of dsDNA may be more relevant in vivo than previously believed.

[1] Department of Physics and Astronomy, and LaserLaB Amsterdam, Vrije Universiteit Amsterdam, De Boelelaan 1081, 1081 HV Amsterdam, The Netherlands. [2] Center for Chromosome Stability and Center for Healthy Aging, Department of Cellular and Molecular Medicine, University of Copenhagen, Blegdamsvej 3B, 2200 Copenhagen N, Denmark. [3] Present address: Institute of Structural and Molecular Biology, University College London, Gower Street, London WC1E 6BT, UK. [4] These authors contributed equally: Julia A. M. Bakx, Andreas S. Biebricher, Graeme A. King. [5] These authors jointly supervised this work: Gijs J. L. Wuite, Erwin J. G. Peterman. ✉email: g.j.l.wuite@vu.nl; e.j.g.peterman@vu.nl

Topoisomerase enzymes are vital for regulating the topology of DNA in vivo. The type 1A class of topoisomerases can both relax supercoils and resolve (i.e., decatenate) entangled DNA intermediates arising from DNA replication and recombination[1–3]. These enzymes contain a toroidal core structure that is highly conserved in all kingdoms of life. The catalytic activity of type 1A topoisomerases relies on the presence of a single-stranded (ss)DNA binding site, and involves the following key steps (Fig. 1)[4–7]. First, the enzyme cleaves the ssDNA by forming a covalent phosphotyrosyl bond with the 5′ end of the break and a tight, but non-covalent, connection to the 3′ end. This yields an enzyme-mediated bridge that serves as a 'gate' in the ssDNA. Next, a conformational rearrangement occurs that results in the opening of the protein-ssDNA gate. This allows a second DNA segment, generally referred to as the transported (T)-DNA, to enter into the central cavity of the enzyme. After T-DNA binding, the protein-ssDNA gate closes and the ssDNA backbone is re-ligated. Each completed cycle thus results in a change of ± one linking number, without the need for ATP[8–11].

Most of our current knowledge of type 1A topoisomerases is derived from studies of the two *Escherichia coli* type 1A enzymes *Ec*TopoI and *Ec*TopoIII (also designated *Ec*Topo1 and *Ec*Topo3, respectively). While both of these enzymes can (de)catenate DNA strands and relax supercoiled DNA[7,12,13], the (de)catenation efficiency of *Ec*TopoIII is considerably higher than that of *Ec*TopoI[12,14–17]. X-ray crystal structures have revealed that, in contrast to *Ec*TopoI, *Ec*TopoIII contains a short looped structure at the base of the central cavity, termed the 'decatenation loop', deletion of which drastically reduces the (de)catenation activity[4,5]. Recent single-molecule studies using magnetic tweezers have shed more detailed insight into the catalytic cycle of these enzymes. For example, Gunn et al. revealed that *Ec*TopoI undergoes multiple conformational changes before successful

strand passage occurs[18]. Furthermore, Mills et al. provided the first direct observation of gate opening for *Ec*TopoI and *Ec*TopoIII, and measured the size of the open gate to be ~5–6 nm[19].

In contrast to prokaryotes, most eukaryotic type 1A topoisomerases require one or more protein co-factors for proper catalytic activity. In budding yeast, for example, *Sc*TopoIII forms a complex with Rmi1 (RecQ-mediated genome instability protein 1)[20–22], while the human homologue, *Hs*TopoIIIα forms a stable complex with co-factors RMI1 and RMI2 (referred to as TRR). In both complexes, the presence of Rmi1/RMI1 has been shown to greatly enhance the (de)catenation activity of the core topoisomerase[20–24]. This has been attributed to the fact that the Rmi1/RMI1 co-factor introduces a looped structure into the central cavity of *Sc*TopoIII/*Hs*TopIIIα, respectively, that resembles the decatenation loop in *Ec*TopoIII[25].

In vivo, those type 1A topoisomerases that have strong (de) catenation activities (such as *Ec*TopoIII, *Sc*TopoIII-Rmi1, and TRR) typically work in concert with helicases of the RecQ family[26–31]. The RecQ family members are highly conserved from prokaryotes (RecQ) to yeast (Sgs1) and humans (BLM). In the dissolution of double Holliday junctions, for example, the ATP-dependent activity of the RecQ helicase ensures branch migration, while the relevant type 1A topoisomerase unlinks the structure into its constituent DNA components[25,31,32]. A similar mechanism has been proposed for the decatenation of hemicatenated DNA that can arise from late replication intermediates[20,25,28,33]. The importance of the RecQ-family helicases for the activity of TopoIII-like topoisomerases is emphasized by the fact that Sgs1/BLM binds directly to both *Sc*TopoIII/*Hs*TopoIIIα[23,29,34] and Rmi1/RMI1[23,24,34,35]. It has also been found that Sgs1/BLM has a stimulatory effect on *Sc*TopoIII-Rmi1/*Hs*TRR decatenation, independent of its unwinding activity[20,22,26,33,36].

Although the general steps of the catalytic cycle of type 1A topoisomerases are known, the mechanisms underlying each step, and how they might be regulated, are still not fully understood. Here, we present a single-molecule approach, based on combined optical tweezers and fluorescence imaging, which has allowed us to obtain valuable insight into distinct steps of the catalytic cycle of TRR. We have directly measured TRR-mediated gate opening of ssDNA, and determined the size of the open gate to be 8.5 ± 3.8 nm (mean ± SD). Furthermore, we have directly visualized the binding of double-stranded (ds)T-DNA and ssT-DNA to the open gate, followed by successful catenation. We also reveal that dsT-DNA binding increases the size of the open gate, whereas the presence of BLM drastically alters the flexibility of the open gate. These results demonstrate how co-factors can regulate the gate opening mechanics of TRR-ssDNA, and suggest that dsT-DNA transfer might play a greater role in vivo than previously thought.

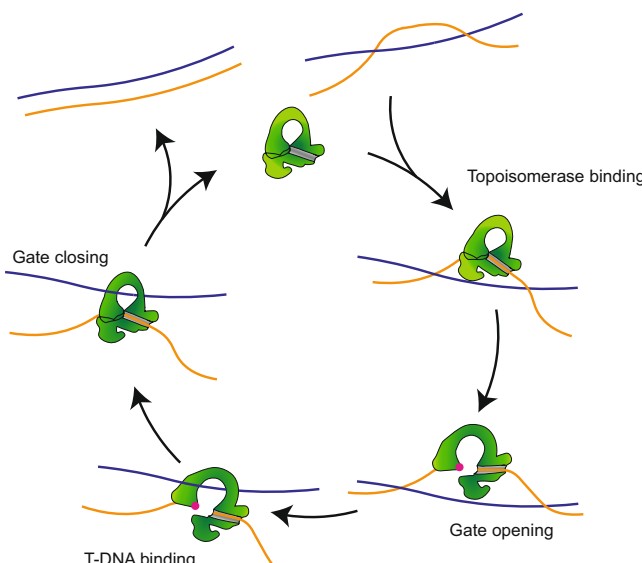

**Fig. 1 Schematic representation of the catalytic cycle of type 1A topoisomerases.** First (top right), the enzyme (green) binds in a closed configuration to a region of ssDNA (orange). The enzyme cleaves the ssDNA backbone upon binding and forms a protein-ssDNA gate (red dot indicates the catalytic site, which forms a covalent bond with the DNA backbone). The protein subsequently undergoes a conformational rearrangement that opens the gate. This allows an adjacent strand of T-DNA (blue) to bind to the protein cavity. Finally, the enzyme-ssDNA gate closes and the ssDNA backbone is re-ligated. For two entwined DNA strands, this cycle results in (de)catenation of the two strands.

## Results

**Direct observation of TRR gate opening on ssDNA.** For all of our experiments, we employed combined dual-trap optical tweezers and multi-colour fluorescence microscopy in a multi-channel fluidic system (Supplementary Fig. 1)[28]. The latter is of particular importance as it allowed us to manipulate a single DNA molecule and incubate it in up to four different buffer and/or protein solutions ("Methods" and Supplementary Table 1). We first used this approach to characterize the mechanical effect of TRR binding to an ssDNA substrate (Fig. 2). To this end, an ssDNA molecule was generated by force-induced melting of lambda-phage (λ) dsDNA (48,502 bp)[37] and then incubated in a solution of TRR fluorescently labelled with mCherry ("Methods"). Successful binding of multiple TRR-mCherry proteins to the tethered ssDNA

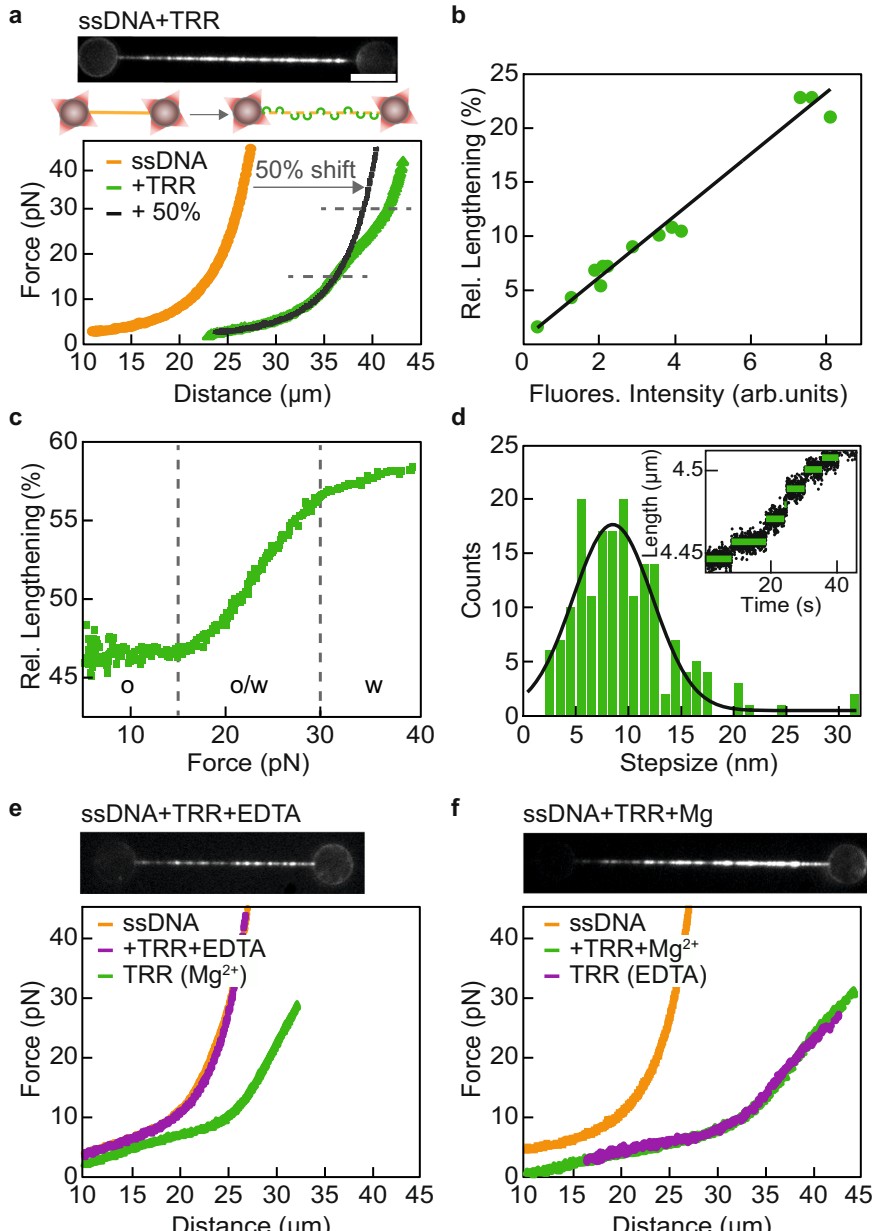

**Fig. 2 Direct observation of TRR gate opening on ssDNA. a** (Top) Representative mCherry fluorescence image of TRR-ssDNA. Scale bar represents 5 μm, and applies to all snapshots. (Bottom) Representative FD-curves of bare ssDNA (orange) and TRR-ssDNA (green). The black line shows the FD-curve of ssDNA shifted to a 50% longer contour length, while the dashed grey lines highlight the shoulder in the FD-curve which signifies an additional, force-induced length increase from ~15 to 30 pN. The schematic representation depicts the length increase of ssDNA due to TRR binding and gate opening. Representative data are shown from at least 30 independent measurements. **b** Relative (rel.) lengthening of ssDNA as a function of mCherry fluorescence (fluores.) intensity from bound TRR, based on at least 10 independent experiments (green), together with a linear fit to the data (black line). **c** 'Subtraction plot' showing the relative lengthening for TRR-ssDNA compared to bare ssDNA as a function of force, based on the data shown in panel (**a**). Dashed grey lines highlight the transition (o/w) between the open (o) and the widened (w) states of the TRR-ssDNA gate. **d** Step-size distribution of single gate opening events (green) following TRR-induced cleavage of ssDNA, fitted to a single Gaussian function (black line, based on $N = 176$; average step size 8.5 ± 3.8 nm; mean ± SD). Inset: Representative length–time trace recorded for ssDNA in the presence of a low concentration (~1 nM) of TRR, measured at a constant force of 15 pN (from six independent measurements). Raw data (black) are shown, together with steps fitted using a step-fitting algorithm (green). **e** (Top) Representative mCherry fluorescence image (from $N \geq 10$) following incubation of a tethered ssDNA substrate in TRR in $Mg^{2+}$-deficient buffer. (Bottom) Representative FD-curves (from $N \geq 10$) of bare ssDNA (orange), TRR-ssDNA in the absence of $Mg^{2+}$ (purple), and the same TRR-ssDNA molecule after incubation in $Mg^{2+}$-containing buffer (green). **f** (Top) Representative mCherry fluorescence image (from $N \geq 15$) following incubation of a tethered ssDNA substrate in TRR in standard buffer (i.e., containing $Mg^{2+}$). (Bottom) Representative FD-curves (from $N \geq 15$) of bare ssDNA (orange), TRR-ssDNA in the presence of $Mg^{2+}$ (green), and the same TRR-ssDNA after incubation in $Mg^{2+}$-deficient buffer (purple). Source data are provided as a Source Data file.

was verified by fluorescence microscopy, after the molecule had been moved into a protein-free buffer channel (Fig. 2a and Supplementary Data 1). No dissociation of TRR from ssDNA was observed under these conditions on our measurement timescales (<10 min). Strikingly, the force–distance (FD) curve of TRR-coated ssDNA (denoted TRR-ssDNA) showed a substantial, and highly reproducible, length increase of up to 50% compared to that of bare ssDNA (Fig. 2a, Supplementary Fig. 2a, and Supplementary Data 1). The length increase was the same, irrespective of whether the TRR was fluorescently-labelled or not (Supplementary Fig. 2b). We attribute this length change to multiple TRR-induced cleavages of the ssDNA backbone and subsequent opening of each protein-mediated ssDNA gate (Supplementary Note 1). This conclusion is supported by the following three observations. First, no length increase of ssDNA was observed for a mutant of TRR (TRR-Y337F) that can bind to but cannot cleave, the ssDNA (Supplementary Fig. 2c). Second, for forces up to ~15 pN, the FD-curve of TRR-ssDNA overlays well with an FD-curve of bare ssDNA shifted to longer extension (Fig. 2a). Finally, the length increase induced by TRR scales linearly with the number of TRR proteins bound to the ssDNA, as determined by quantification of the fluorescence intensity (Fig. 2b).

Repeated stretching of the TRR-ssDNA construct resulted in almost identical FD-curves, and only a small hysteresis was observed between the extension and retraction FD-curves (Supplementary Fig. 2d, e). These observations indicate that the increase in ssDNA length induced by the bound TRR equilibrates rapidly (<1 s) even at forces below 5 pN (Supplementary Note 2). Moreover, the TRR-induced lengthening of ssDNA is largely force-independent from 5 to 15 pN. This can be demonstrated most clearly by plotting the difference in length between TRR-ssDNA and bare ssDNA as a function of force, based on the respective FD-curves (see "Methods"), as shown in Fig. 2c and Supplementary Fig. 2f. Note that in these so-called 'subtraction plots', we only consider forces above 5 pN, since it is difficult to determine if length changes below 5 pN are caused by changes of the gate size alone (Supplementary Note 3). The 'subtraction plot' in Fig. 2c also reveals an additional lengthening regime from 15 to 30 pN, which corresponds to an inflection ('shoulder') in the FD-curve (Fig. 2a). Based on these findings, we conclude that most TRR-ssDNA gates are open by 5 pN and that the additional lengthening above ~15 pN reflects a widening of the open gates. While we cannot easily measure TRR-ssDNA lengthening at forces below 5 pN, we nevertheless estimate that a substantial fraction of gates is open at 0 pN (Supplementary Note 4).

If the ssDNA length increase discussed above corresponds to the opening of multiple TRR gates on ssDNA, we would expect to observe a step-wise elongation of the ssDNA at low TRR concentrations due to binding of individual proteins, similar to that reported recently for *Ec*TopoI and *Ec*TopoIII[19] as well as for *Sc*TopoIII[36]. To confirm this, we recorded the change in DNA end-to-end length at constant force (15 pN) at a low protein concentration (<1 nM). In this way, we observed a stepwise lengthening of the ssDNA over time as an increasing number of TRR complexes bound to, cleaved, and opened the ssDNA (Fig. 2d). By extracting the step sizes from many such traces using a step-fitting algorithm (Supplementary Software 1), we obtained a distribution which could be fitted well with a Gaussian function (Fig. 2d), from which we conclude that the open gate size for a single TRR is 8.5 ± 3.8 nm (mean ± standard deviation, SD). This value is comparable to that reported recently for *Sc*TopoIII-Rmi1 (8.2 ± 0.2 nm; mean ± standard error of the mean, SEM)[36] and slightly larger than that for *Ec*TopoI and *Ec*TopoIII (5.9 ± 0.6 and 5.5 ± 0.4 nm (mean ± SD), respectively)[19]. Our data additionally suggest that the size of the open TRR-ssDNA gate is constant for forces from 5 to 15 pN, since the length shift in the FD-curve of

ssDNA due to TRR is constant over this force range (Fig. 2a, c). Although we do not observe that the size of the open gate changes with force above 5 pN, we cannot completely rule out that the gate size is different in the absence of tension.

**ssDNA cleavage, but not gate opening, requires magnesium**. Despite recent advances, the precise role of magnesium cations ($Mg^{2+}$) in the catalytic mechanism of type 1A topoisomerases is still under debate[18,19,38]. We, therefore, applied our FD-curve analysis approach to investigate the influence of $Mg^{2+}$ ions on the interaction of TRR with ssDNA. To this end, we first incubated ssDNA in TRR in the absence of $Mg^{2+}$ (by replacing the $MgCl_2$ in our standard buffer by 1 mM EDTA; see "Methods"). We observed that the corresponding FD-curves did not exhibit a shift in length relative to that of bare ssDNA, even though substantial binding of TRR to ssDNA was observed (Fig. 2e and Supplementary Data 1). This suggests that ssDNA cleavage and/or gate opening of TRR is inhibited in the absence of $Mg^{2+}$. Consistent with this, after re-incubation of the same TRR-bound ssDNA molecule in a buffer containing 2 mM $MgCl_2$, a length shift in the FD-curve was once again observed (Fig. 2e). We note, however, that the magnitude of this length shift was less extensive than that observed in Fig. 2a. This can be explained by a reduced binding of TRR to ssDNA in the absence of $Mg^{2+}$, as evidenced by the concomitant reduction in TRR-mCherry fluorescence on the ssDNA (cf. images in Figs. 2e and 2a). The reduced ssDNA-binding affinity of TRR in the absence of $Mg^{2+}$ is consistent with studies of other type 1A topoisomerases[18,19]. We next performed an alternative experiment, where we moved a TRR-coated ssDNA molecule from a buffer containing 2 mM $MgCl_2$ into a $Mg^{2+}$-deficient buffer (supplemented with 1 mM EDTA). In this case, substantial lengthening was observed both before and after moving to the $Mg^{2+}$-free buffer channel (Fig. 2f and Supplementary Data 1). Together, these results indicate that $Mg^{2+}$ ions are required for backbone-cleavage, but have little or no effect on gate opening.

**Long DNA substrates can stably interact with TRR-ssDNA**. After investigating TRR-ssDNA gate opening, we next sought to capture the following step in the catalytic cycle of TRR, namely the interaction of the open TRR-ssDNA gate with a segment of T-DNA. To this end, we first incubated TRR-ssDNA in a solution of short dsDNA or ssDNA fragments (30 basepairs(bp)/nucleotides(nt), labelled with Atto-647N). However, this did not result in a stable interaction (Supplementary Fig. 3a). This is perhaps not surprising, since the interaction of T-DNA with the central cavity of type 1A topoisomerases has been reported to be transient, and binding of T-DNA has so far only been detected indirectly, using a modified topoisomerase construct[11]. Nevertheless, we occasionally observed that TRR-ssDNA substrates could stably interact with excess λ-dsDNA molecules (tethered to the optically-trapped beads), which are much longer than the short fragments used above (Supplementary Fig. 3b). To investigate the interaction of long dsDNA molecules with TRR-ssDNA under more controlled conditions, we incubated TRR-ssDNA substrates in a solution of long dsDNA molecules (4,361 bp, "Methods"). The binding of these dsDNA molecules to the TRR-ssDNA substrate was visualized by subsequently moving the tethered substrate into a buffer containing fluorescent intercalator dye (SYBR Gold). Under our experimental conditions, we observed extensive and stable binding of long dsDNA to the TRR-ssDNA substrate (Fig. 3a, Supplementary Fig. 3c–f, Supplementary Data 2, and Supplementary Note 5). We posit that extensive binding of long dsDNA molecules is observed because they can make many transient interactions with multiple TRR proteins

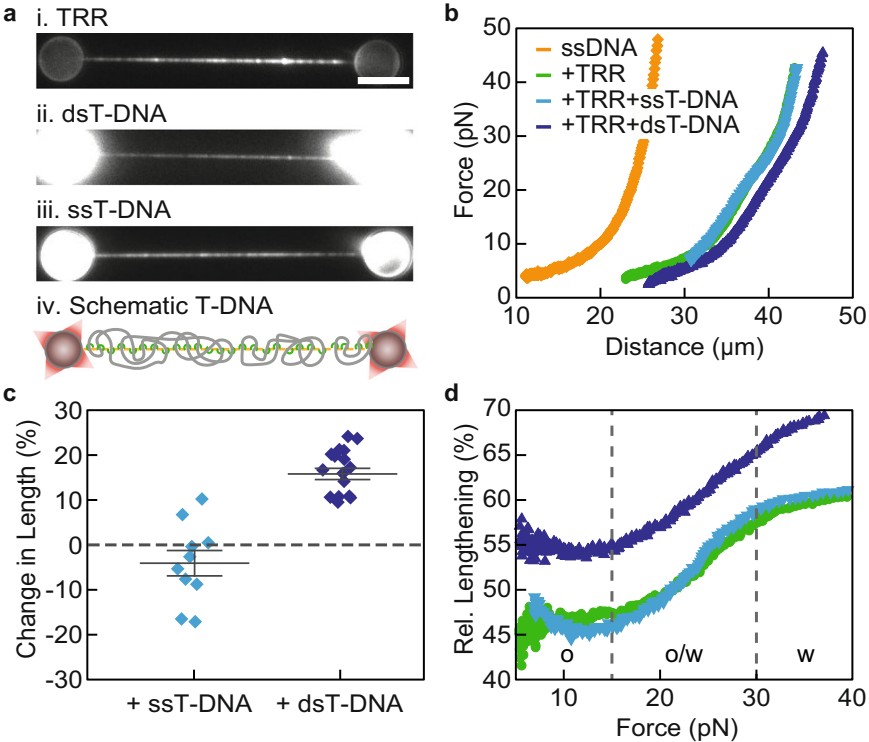

**Fig. 3 Direct observation of ds/ssT-DNA binding to TRR-ssDNA. a** Representative fluorescence images (from $N \geq 10$) of TRR-ssDNA (i) bound by either dsT-DNA (ii) or ssT-DNA (iii). TRR was visualized using mCherry fluorescence, while dsT- and ssT-DNA were stained with intercalator dye. Note that images i and ii correspond to the same substrate (the TRR-mCherry fluorescence image corresponding to the substrate in iii is not shown). A schematic representation of ssDNA coated with TRR and T-DNA is shown underneath (iv). Scale bar represents 5 μm, and applies to all snapshots. **b** Representative FD-curves (from $N \geq 10$) of bare ssDNA (orange), TRR-ssDNA (green) and TRR-ssDNA bound by either dsT-DNA (dark blue) or ssT-DNA (light blue). **c** Change in length of TRR-ssDNA (in the force range from 5 to 15 pN) upon binding of either ssT-DNA (light blue, from $N \geq 10$) or dsT-DNA (dark blue, from $N \geq 10$). Error bars indicate SEM. **d** 'Subtraction plots' showing the relative (rel.) lengthening for TRR-ssDNA (green), and TRR-ssDNA bound by either ssT-DNA (light blue) or dsT-DNA (dark blue) compared to bare ssDNA. Plots were generated using the FD-curves in panel (**b**). Dashed grey lines highlight the transition (o/w) between the open (o) and the widened (w) states of the TRR-ssDNA gate. Source data are provided as a Source Data file.

bound to the tethered ssDNA substrate (Supplementary Note 6). Note that the above observations were independent of whether linear or circular dsDNA molecules were used (Supplementary Fig. 3d, f).

Strikingly, the corresponding FD-curves showed that dsDNA binding to TRR-ssDNA results in a significant length increase relative to that of the TRR-ssDNA substrate (Fig. 3b and Supplementary Data 2). For tethered ssDNA that was fully coated with TRR, the binding of dsDNA induced a constant length increase (relative to TRR-ssDNA) of $16.2 \pm 1.3\%$ (mean ± SEM; $N = 14$) at forces between 5 and 15 pN (Fig. 3c). Aside from this length shift, however, the FD-curve of TRR-ssDNA was largely unaffected by the bound dsDNA (Fig. 3b). For example, the gate widening transition from 15 to 30 pN was still present, as can also be appreciated from the corresponding 'subtraction plot' (Fig. 3d). Note that here, as well as all other 'subtraction plots', the change in length is calculated with respect to that of bare ssDNA. Interestingly, stable dsDNA interactions with TRR-ssDNA were not observed when the intercalator dye was in the same channel as the dsDNA (Supplementary Fig. 4a and Supplementary Movie 1). Since both the protein cavity[5] and cyanine intercalators[39] are known to carry positive charges, we posit that intercalated DNA likely exhibits reduced electrostatic interactions with the protein cavity. Together, these findings support the idea that dsDNA makes direct contact with the central cavity of TRR.

The length increase of TRR-ssDNA due to dsDNA binding could have two explanations: (1) bound dsDNA alters the equilibrium of TRR-ssDNA gate opening; or (2) bound dsDNA

increases the size of the open TRR-ssDNA gate. As explained earlier, the majority of TRR-ssDNA gates are already open at low forces, even in the absence of dsDNA binding. Moreover, if the binding of dsDNA were to alter the TRR-ssDNA gate opening equilibrium, we would expect a significant reduction of the hysteresis in the FD-curve between extension and retraction, which we did not observe (Supplementary Fig. 4b). We, therefore, attribute the increase in length of the TRR-ssDNA substrate when bound by dsDNA to an increase in the gate size. It should be noted, however, that the 16% lengthening deduced here is only a lower bound since we cannot determine if all TRR gates interact with the bound dsDNA molecules.

We next sought to probe the interaction of long ssDNA molecules with TRR-ssDNA. To this end, we repeated the above experiments, but using long circular ssDNA molecules (7,249 nt). To stain the ssDNA, we again used fluorescent intercalator dye, taking advantage of the propensity of ssDNA to form local dsDNA hairpin structures. Figure 3a and Supplementary Fig. 4c show representative fluorescence images, revealing extensive and stable binding of long ssDNA molecules to the TRR-ssDNA substrate, similar to the case for long dsDNA molecules. Despite this strong interaction, the corresponding FD-curve and 'subtraction plot' (recorded in the absence of intercalator dye) showed no increase in end-to-end length relative to that of TRR-ssDNA (Fig. 3b–d). These observations indicate that, in contrast to dsDNA, ssDNA molecules do not alter the TRR-ssDNA gate size. Furthermore, they disprove the notion that hairpin structures (i.e., local dsDNA sections) play a substantial role in the observed

ssDNA binding to the TRR-ssDNA substrate, since that would result in an increase in the gate size (as identified above for dsDNA binding). We subsequently refer to the interactions of T-DNA with the TRR cavity as dsT-TRR or ssT-TRR, where dsT and ssT represent the bound T-DNA.

Our observation that dsT-DNA binds strongly to TRR-ssDNA raises the question of whether the affinity of dsT-DNA is high enough to compete with that of ssT-DNA. To investigate this, we incubated TRR-ssDNA in an equimolar mixture of dsT- and ssT-DNA. Since intercalators effectively stain both dsT- and ssT-DNA, they cannot be used to differentiate their respective binding to TRR-ssDNA under the conditions used above. Therefore, we used a modified staining protocol that permitted us to exclusively stain dsT-DNA with intercalators, while bound ssT-DNA could be visualized by a 2nd staining step with free TRR (Supplementary Note 7). Using this approach, substantial binding of both dsT- and ssT-DNA to the TRR-ssDNA substrate was detected (Supplementary Fig. 4d). We conclude, therefore, that dsT-DNA has sufficient affinity for TRR-ssDNA to compete with ssT-DNA. In addition, we note that dsT-DNA binding to TRR-ssDNA can be observed at very low forces (<2 pN), indicating that elevated force is not required to induce T-DNA binding (Supplementary Note 8 and Supplementary Fig. 4e).

**Direct visualization of ds/ssT-DNA catenation with ssDNA.** Although our results indicate that the TRR-ssDNA gate opening equilibrium lies towards the open state, this does not preclude the existence of a dynamic equilibrium that would allow gates to close in a transient manner. If this were to happen, we would expect that circular T-DNA would become catenated with the tethered λ-ssDNA molecule. The data shown in Fig. 3, nevertheless, do not allow us to differentiate between the binding of T-DNA to TRR-ssDNA and TRR-mediated catenation of T-DNA around ssDNA. We, therefore, exploited the known effect of high salt (500 mM NaCl) to induce dissociation of TRR from the λ-ssDNA substrate[29]. We expected that if circular T-DNA molecules (either ds or ss) were catenated with the tethered λ-ssDNA by TRR, these plasmids should remain connected to the λ-ssDNA after salt-induced TRR dissociation. We first confirmed that incubation of TRR-ssDNA in a high-salt buffer (standard buffer containing 500 mM NaCl, "Methods") induces the majority of TRR (71 ± 6%, mean ± SEM; $N = 9$) to dissociate (Fig. 4a, b and Supplementary Data 3). Note that, after moving to the high-salt channel, the tension on the tethered λ-ssDNA was reduced (<1 pN) in order to increase the probability of (transient) TRR-ssDNA gate closure, and thus TRR unbinding ("Methods"). We next repeated the above experiments, but this time we incubated the TRR-ssDNA substrate in a solution of circular dsT-DNA molecules prior to moving the substrate to a highsalt buffer containing intercalator dye. We again observed significant TRR dissociation in high-salt buffer (Supplementary Fig. 5a), but also a strong fluorescence signal from intercalated dsT-DNA connected to the λ-ssDNA substrate. These bound dsT-DNA molecules were highly mobile on the λ-ssDNA, as demonstrated by applying buffer flow to the substrate (Fig. 4c, Supplementary Movie 2 and Supplementary Data 3). We conclude, therefore, that these dsT-DNA molecules had been catenated with the λ-ssDNA. This finding was substantiated by control experiments using linearized, rather than circular, dsT-DNA. Unlike circular dsT-DNA, linear dsT-DNA cannot be topologically entrapped on the tethered ssDNA upon catenation by TRR and will therefore diffuse away. Accordingly, when using the linearized plasmid, substantially less dsT-DNA remained bound to the ssDNA molecule after moving to high-salt buffer (cf. Fig. 4d, e). Moreover, the small residual fraction of linear dsT-DNA showed no diffusion (Fig. 4e) and was therefore likely bound to remaining

TRR complexes that had not dissociated from the ssDNA. Note that successful catenation of circular dsT-DNA could also be demonstrated under physiological salt conditions, using a lower TRR coating to enable diffusion of the catenated molecules (Supplementary Fig. 5b and Supplementary Movie 3). This observation confirms that, although diffusion is more extensive in high-salt conditions, high salt is not a requirement for catenation.

We subsequently tested whether ssT-DNA molecules could also be catenated by TRR. To this end, we repeated the above experiments, but this time incubated the TRR-ssDNA substrate in a channel containing circular ssT-DNA, rather than dsT-DNA. After moving to high-salt buffer, we observed a strong coating of ssT-DNA on the λ-ssDNA, despite substantial TRR dissociation (Supplementary Fig. 5c). Since we independently confirmed that long ssDNA molecules in solution cannot interact stably with the tethered λ-ssDNA substrate in the absence of TRR (Supplementary Fig. 5d), we conclude that catenation of ssT-DNA by TRR had taken place. Interestingly, we did not observe significant diffusion of the catenated ssT-DNA molecules on the λ-ssDNA substrate, in contrast to the case for circular dsT-DNA (Fig. 4c). However, this is most likely due to friction between the λ-ssDNA template and the catenated ssDNA plasmid, either as a result of local base-pairing interactions and/or steric hindrance associated with hairpin structures within the ssDNA plasmid.

**EcTopoI can efficiently catenate ssT-DNA but not dsT-DNA.** Our next goal was to determine if the T-DNA binding behaviour observed above is conserved in other type 1A topoisomerases. To this end, we repeated these T-DNA-binding experiments using EcTopoI instead of TRR. Comparison of TRR with EcTopoI is particularly relevant for two reasons. First, EcTopoI is the prototype of the 'TopoI' sub-family of type 1A topisomerases, which are more efficient at supercoil relaxation than (de)catenation[12–14], whereas TRR is primarily considered to be a (de)catenase[28,30,40]. Second, the catalytic core of EcTopoI shows strong sequence similarity to that of HsTopoIIIα[3,25]. We first demonstrated that binding of unlabelled EcTopoI to λ-ssDNA (EcTopoI-ssDNA) yields an FD-curve with a substantial shift in length relative to that of bare ssDNA, as well as a shoulder at ~20 pN (Fig. 5a and Supplementary Data 4). These features are broadly similar to those observed for TRR-ssDNA, where the length shift is attributed to gate opening, and the shoulder is due to gate widening. However, the corresponding 'subtraction plot', showing the length of EcTopoI-ssDNA relative to that of bare ssDNA as a function of force, reveals significant differences with that of TRR-ssDNA (Fig. 5b). First, substantial lengthening of ssDNA due to bound EcTopoI only occurred at forces above ~10 pN, consistent with results from recent magnetic tweezers studies[19]. Second, EcTopoI resulted in a much lower maximum lengthening (20–25%) of ssDNA compared to TRR (50%). We attribute this to a reduced number of EcTopoI proteins bound to the ssDNA compared with TRR, due to the fact that the footprint of EcTopoI on ssDNA is much larger (~40 nt)[41] than TRR (~20 nt)[28,42]. Finally, the gate widening transition in the case of EcTopoI-ssDNA was much more extensive than for TRR-ssDNA. We also probed how $Mg^{2+}$ affects backbone cleavage and gate opening, respectively, by EcTopoI. Similar to the case for TRR, $Mg^{2+}$ was required for backbone cleavage, but did not appear to influence gate opening significantly (Supplementary Fig. 6a, b).

Next, we investigated the binding of dsT-DNA to EcTopoI-ssDNA. To this end, EcTopoI-ssDNA was incubated first in a solution of long dsT-DNA molecules and then in a solution of intercalator dye, similar to the approach used in Fig. 3a. We observed that the coating of dsT-DNA on EcTopoI-ssDNA was much reduced compared to TRR-ssDNA (Fig. 5c and

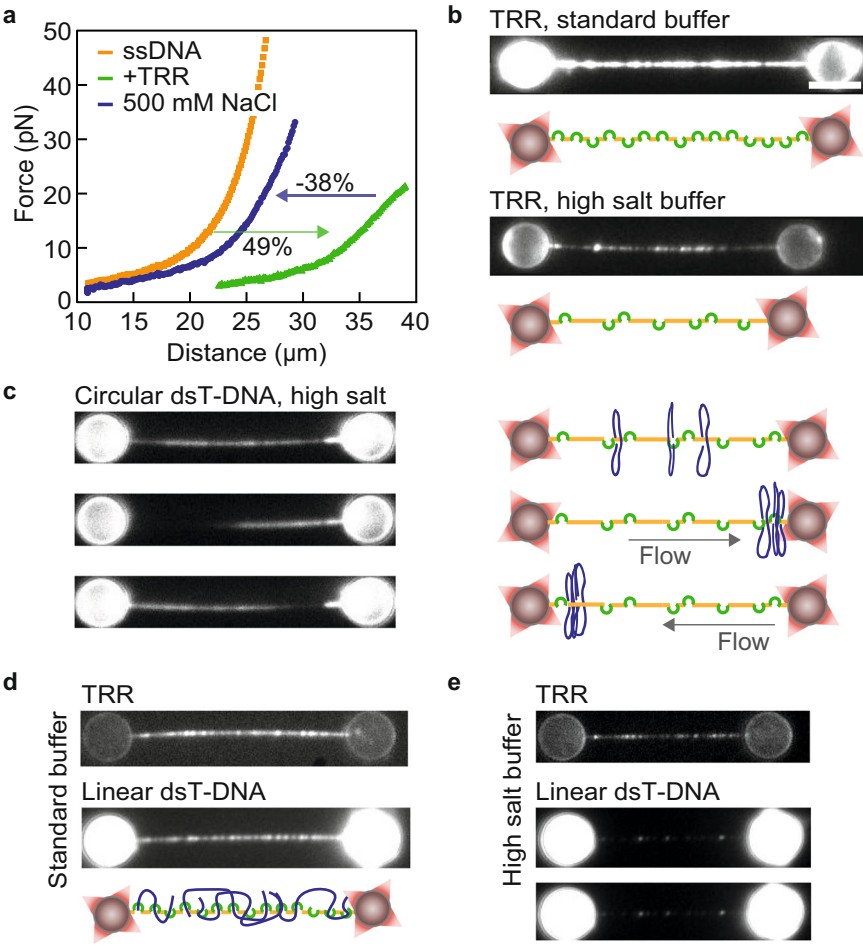

**Fig. 4 Catenation of circular dsDNA around tethered ssDNA by TRR. a** Representative FD-curves (from $N \geq 20$) of TRR-ssDNA before (green) and after (blue) incubation in high-salt buffer. For reference, the FD-curve of bare ssDNA is shown in orange. Arrows depict the changes in end-to-end distance: TRR binding in standard buffer results in ~49% lengthening of the ssDNA, which was reversed by ~−38% after moving the substrate into high-salt buffer due to TRR unbinding. **b** Representative mCherry fluorescence images (from $N \geq 20$) of TRR-ssDNA before (top) and after (bottom) incubation in high-salt buffer. The scale bar represents 5 μm, and applies to all snapshots. **c** Representative intercalator fluorescence images (left; from $N \geq 10$) and schematic representations (right) of circular dsT-DNA catenated around tethered TRR-ssDNA after incubation in high-salt buffer under different buffer flow conditions. The applied flow helps visualize the high mobility of the dsT-DNA. **d** Representative fluorescence images (from $N = 3$) of TRR-ssDNA (mCherry fluorescence, top) bound by linear dsT-DNA (intercalator staining, centre), recorded in standard buffer. A schematic representation of the interaction is also shown (bottom). **e** Representative fluorescence images (from $N = 3$) of TRR-ssDNA (mCherry fluorescence, top) bound by residual linear dsT-DNA (intercalator staining, bottom two images), recorded in high-salt buffer (same molecule as in panel (**d**)). The two bottom images were obtained several seconds apart and show a similar (albeit weak) fluorescence intensity pattern, suggesting that the dsT-DNA was bound to residual TRR that remained on the ssDNA. Note that when incubating in high-salt buffer, the tension was reduced to <1 pN in order promote protein unbinding. Source data are provided as a Source Data file.

Supplementary Data 4), suggesting that dsT-DNA has a lower affinity for EcTopoI-ssDNA than for TRR-ssDNA. Correspondingly, we detected far fewer catenation events of circular dsT-DNA for EcTopoI-ssDNA than for TRR-ssDNA after incubating the substrates in high-salt buffer (Fig. 5d and Supplementary Data 4). We additionally note that the presence of dsT-DNA did not alter the FD-curves of EcTopoI-ssDNA, in contrast to TRR-ssDNA (Fig. 5e). This might either reflect the lower binding affinity of dsT-DNA to EcTopoI-ssDNA compared to TRR-ssDNA, or suggest that dsT-DNA has no effect on the gate size of EcTopoI-ssDNA.

We also probed the binding of long circular ssT-DNA molecules to EcTopoI-ssDNA using the same approach as above. These experiments revealed that long ssT-DNA interacts with EcTopoI-ssDNA in a similar manner as it does with TRR-ssDNA; thus, we observed extensive and stable binding of ssT-DNA to EcTopoI-ssDNA (Fig. 5f and Supplementary Data 4), even after protein dissociation induced by incubation in high-salt buffer

(Fig. 5g and Supplementary Data 4). Similar to the case for TRR-ssDNA, the extensive binding of long ssT-DNA molecules to EcTopoI-ssDNA had no influence on the corresponding FD-curve (Fig. 5e). Taken together, these findings suggest that, while long ssT-DNA molecules can bind to both EcTopoI-ssDNA and TRR-ssDNA, leading to catenation, long dsT-DNA molecules are much more efficiently bound to and catenated by TRR-ssDNA than by EcTopoI-ssDNA.

**BLM alters the mechanical properties of the TRR-ssDNA gate.** Since BLM has been shown to stimulate TRR activity[20,26,33,36,43] we next investigated whether BLM would also influence TRR-ssDNA gate opening. We first incubated λ-ssDNA in an equimolar solution of TRR-mCherry and SNAP649-labelled BLM (together referred to as BTRR). As expected, we observed strong binding and co-localization of both proteins on λ-ssDNA (Fig. 6a)[28]. However,

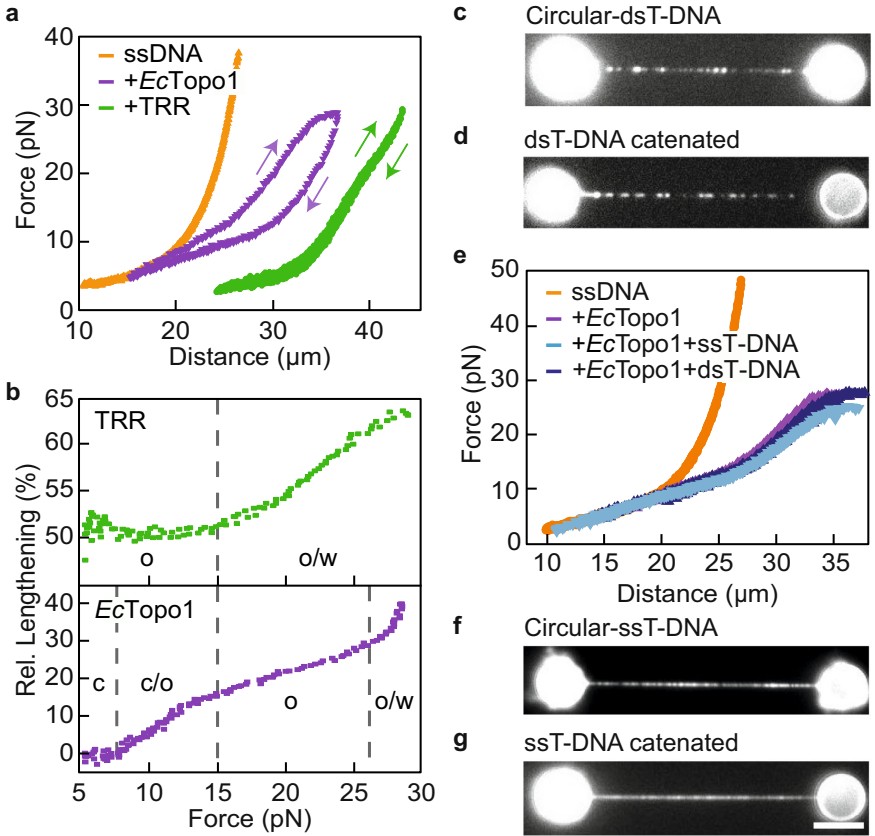

**Fig. 5 Binding and catenation of dsT- and ssT-DNA to tethered ssDNA by EcTopoI. a** Representative FD-curves (from N ≥ 30) of bare ssDNA (orange), TRR-ssDNA (green) and EcTopoI-ssDNA (purple). Upwards and downwards arrows indicate extension and retraction curves, respectively. **b** 'Subtraction plots' showing the relative (rel.) lengthening for TRR-coated ssDNA (top, green) and EcTopoI-coated ssDNA (bottom, purple) compared to bare ssDNA. These plots were generated from the FD-curves shown in panel (**a**). Dashed grey lines highlight the transitions (c/o, o/w) between the closed (c), open (o) and widened (w) states of the topoisomerase-ssDNA gate. **c** Representative intercalator fluorescence image (from N ≥ 5) of circular dsT-DNA bound to EcTopoI-ssDNA in standard buffer. **d** Representative intercalator fluorescence image (from N ≥ 5) of circular dsT-DNA obtained after moving the substrate shown in panel (**c**) into high-salt buffer. **e** Representative FD-curves (from N ≥ 5) of bare ssDNA (orange), EcTopoI-ssDNA (purple) and EcTopoI-ssDNA in the presence of either ssT-DNA (light blue) or dsT-DNA (dark blue) in standard buffer. **f** Representative intercalator fluorescence image (from N = 4) of circular ssT-DNA bound to EcTopoI-ssDNA in standard buffer. **g** Representative intercalator fluorescence image (from N = 4) of circular ssT-DNA catenated around ssDNA obtained after moving the substrate shown in panel (**f**) into high-salt buffer. Scale bar represents 5 μm, and applies to all snapshots. Source data are provided as a Source Data file.

when we compared the FD-curve of BTRR-ssDNA to that of TRR-ssDNA, we observed notable differences (Fig. 6b and Supplementary Data 5). First, it was not possible to overlay the FD-curve of BTRR-ssDNA with that of either bare ssDNA or TRR-ssDNA by a shift in extension alone. Second, the characteristic shoulder observed in the FD-curves of TRR-ssDNA at ~20 pN was not observed for BTRR-ssDNA. Third, although the end-to-end length of BTRR-ssDNA was longer than that of bare ssDNA, the construct was significantly shorter than TRR-ssDNA at forces >5 pN (Fig. 6b). This can also be appreciated by comparing the 'subtraction plots', showing the difference in length between TRR-ssDNA and BTRR-ssDNA, respectively, and bare ssDNA (Fig. 6c). These plots indicate that the presence of BLM perturbs the mechanical properties of the TRR-ssDNA gate such that the distinction between gate opening and gate widening is no longer discernible. Since the TRR-mCherry fluorescence intensity was even slightly higher for BTRR than for TRR alone (Supplementary Fig. 7a), the altered mechanical properties in the presence of BLM cannot be accounted for by reduced TRR binding. Furthermore, we observed the same effect of BLM independent of whether we incubated the ssDNA in TRR and BLM together or whether the TRR-ssDNA was incubated in BLM separately (Supplementary Fig. 7b). This suggests that the altered mechanical properties are

also not due to BLM hindering the interaction of TRR with ssDNA. As an additional control, we also recorded FD-curves of BLM-ssDNA (i.e., BLM bound to λ-ssDNA), which revealed only minor changes compared to the FD-curve of bare ssDNA (Supplementary Fig. 7c). The apparent shortening at forces >5 pN induced by BLM binding to TRR-ssDNA could either be caused by 1) BLM altering the equilibrium of TRR-ssDNA gate opening, or 2) BLM reducing the size of the open TRR-ssDNA gates. If BLM were to shift the equilibrium towards the closed gate state, this would result in a greater number of closed gates at low force compared with TRR-ssDNA. In that case, we would expect that force-induced opening of these closed gates would lead to a larger hysteresis between the forward and backward FD-curves of BTRR-ssDNA compared with those of TRR-ssDNA. Since we did not observe an increase in the hysteresis (Supplementary Fig. 7d), we conclude that the reduction in length of TRR-ssDNA upon BLM binding is most likely due to a decrease of the gate size.

**Interaction of dsT-DNA with BTRR-ssDNA.** Having established that BLM alters the mechanical properties of TRR-ssDNA, we next investigated whether the presence of BLM also affects T-DNA binding. To this end, we incubated

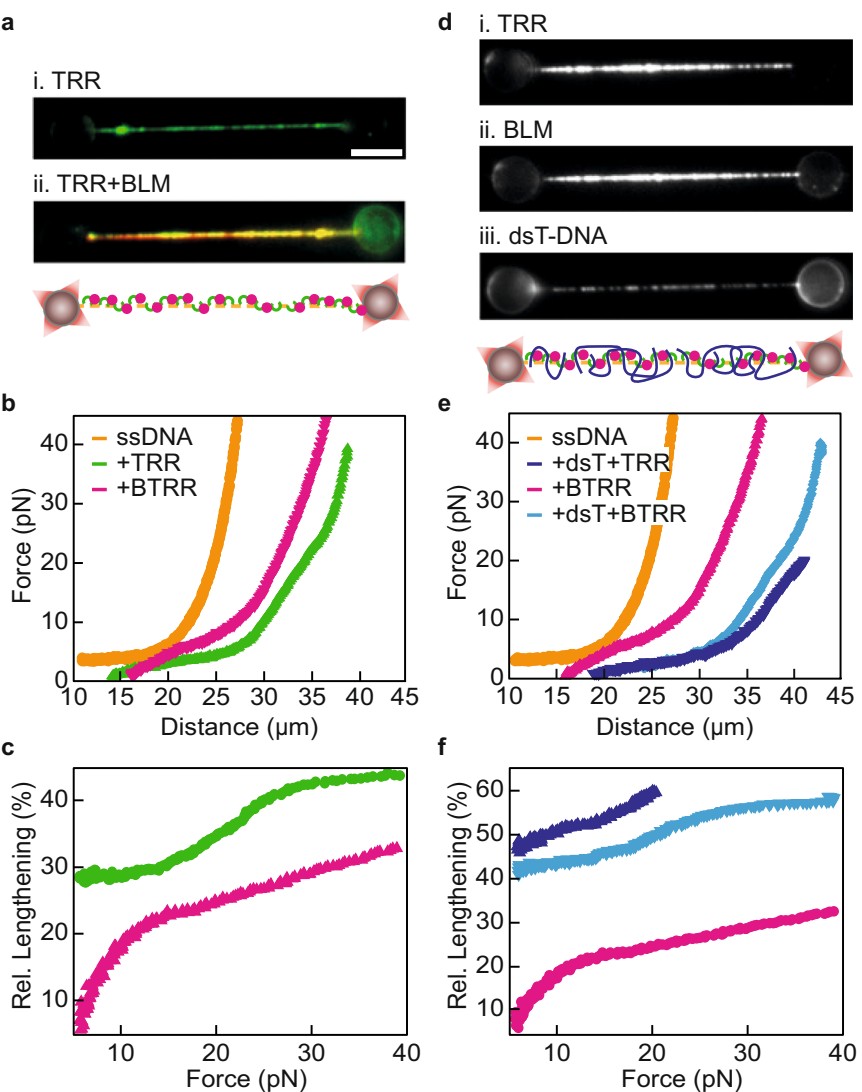

**Fig. 6 BLM alters the flexibility of the TRR-ssDNA gate. a** Representative fluorescence images (from $N \geq 10$) of TRR-ssDNA (i) and BTRR-ssDNA (ii), visualized using mCherry/SNAP649 fluorescence (green/red), respectively. A schematic representation of BTRR complexes bound to ssDNA is shown underneath. The scale bar represents 5 μm, and applies to all snapshots. **b** Representative FD-curves (from $N \geq 10$) of TRR-ssDNA (green) and BTRR-ssDNA (pink). For reference, the FD-curve of bare ssDNA is shown in orange. **c** 'Subtraction plots' showing the relative (rel.) lengthening for TRR-ssDNA (green) and BTRR-ssDNA (pink) compared to bare ssDNA. Plots were generated from the FD-curves shown in panel (**b**). **d** Representative fluorescence images (from $N \geq 5$) of BTRR-ssDNA bound by dsT-DNA. TRR and BLM were visualized using mCherry (i) and SNAP649 (ii) fluorescence, while dsT-DNA was stained with intercalator dye (iii). A schematic representation is shown below. **e** Representative FD-curves (from $N \geq 5$) of BTRR-ssDNA in the absence (pink) and presence (light blue) of dsT-DNA. For reference, the FD-curves of bare ssDNA (orange) and TRR-ssDNA bound by dsT-DNA (dark blue) are also shown. **f** 'Subtraction plots' showing the relative lengthening for dsT-coated TRR-ssDNA (dark blue), BTRR-ssDNA (pink) and dsT-coated BTRR-ssDNA (light blue) compared to bare ssDNA. Plots were generated from the data shown in panel (**e**). Source data are provided as a Source Data file.

λ-ssDNA in a solution containing BLM, TRR and dsT-DNA, and subsequently moved the substrate into an intercalator-containing channel in order to visualize bound dsT-DNA. This experiment confirmed that dsT-DNA binds stably to BTRR-ssDNA (Fig. 6d and Supplementary Data 5). Moreover, the binding of dsT-DNA to BTRR-ssDNA induced a significant increase in the length of the substrate (Fig. 6e, f and Supplementary Data 5) and more generally lacked the three characteristics observed for BTRR-ssDNA FD-curves explained above. Note that these FD-curves were recorded inside the dsT-DNA channel. However, when BTRR-ssDNA was incubated in dsT-DNA and then an FD-curve was recorded outside the dsT-DNA channel (similar to the conditions used for Fig. 6d), the dsT-induced length increase of BTRR-ssDNA

was less extensive (Supplementary Fig. 7e). This suggests that dsT-DNA has a lower affinity for BTRR-ssDNA than for TRR-ssDNA, where such an effect was not observed. Note that the above experiments were conducted at a much lower («10-fold) dsT-DNA concentration compared to that used for studies of dsT-TRR-ssDNA (Fig. 3). Interestingly, when these experiments were performed at a dsT-DNA concentration comparable to that used in Fig. 3, we detected a substantial reduction in the fraction of bound BLM (Supplementary Fig. 7f). Taken together, these findings suggest that BLM and dsT-DNA act antagonistically on TRR-ssDNA. A schematic representation of the influence of different binding partners (dsT-DNA, sst-DNA, and BLM) on the mechanical properties of the TRR-ssDNA gate is provided in Fig. 7a.

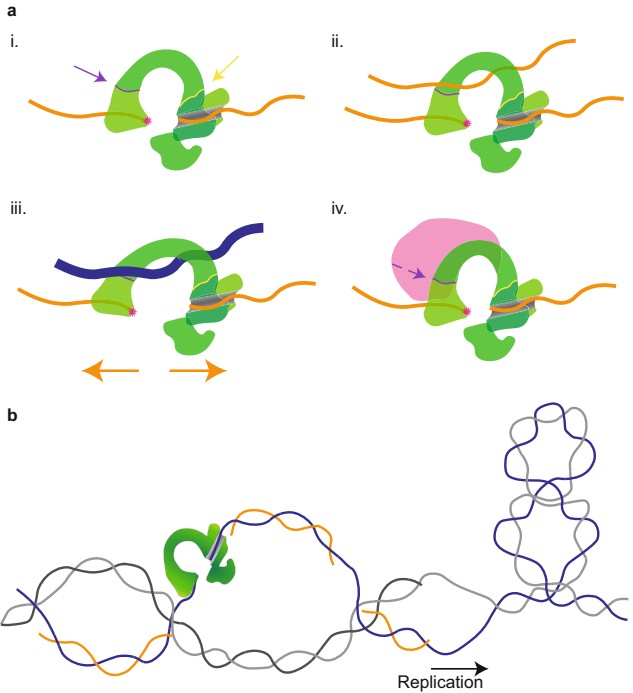

**Fig. 7 Schematic representation of TRR-ssDNA gate plasticity and its potential relevance in vivo. a** Effects of co-factors on TRR-ssDNA gate mechanics. i Upon binding to ssDNA (orange), TRR (green) pivots to create an open gate in the TRR-ssDNA complex. The purple line and arrow highlight the location of a primary hinge in the enzyme that facilitates gate opening. An additional potential hinge is indicated by the yellow line and arrow. ii Binding of ssT-DNA (second orange strand) to the open TRR-ssDNA gate does not influence the gate size. iii Binding of dsT-DNA (blue) to the open TRR-ssDNA gate induces an increase of the gate size by at least ~16%. iv Proposed interaction of BLM (pink) with TRR in the BTRR complex. BLM is hypothesized to alter the overall flexibility of the gate by making contact near the primary hinge (purple line and arrow) of the topoisomerase. **b** Schematic representation of potential in vivo activity of TRR on pre-catenanes behind the replication fork (arrow indicates the direction of replication). TRR (green) may bind to ssDNA on the lagging strand template (blue), between Okasaki fragments (orange), and mediate transfer of the leading strand dsDNA (grey).

## Discussion

Here, we have developed a single-molecule strategy to probe key steps of the catalytic cycle of TRR. Our experimental approach, which can be applied more generally to study type 1A topoisomerases, exploits two unique features. The first is the use of FD-curve analysis to measure the average gate opening of many enzymes bound to a single ssDNA molecule. This is powerful because it provides a sensitive means to measure subtle conformational changes in protein structure that would be difficult to achieve based on single gate opening events (which would require a large number of events to achieve similar statistics). Using this approach, we revealed that most TRR-ssDNA gates are open at forces from at least 5 pN. Moreover, we determined that dsT-DNA binding increases the size of the open gate by a minimum of ~16%, although the overall mechanical properties of gate opening are unchanged. Given that we observe no force dependence of the gate size from 5 to 15 pN, and that previous simulations of EcTopoIII indicate a distinct opening transition in the absence of tension[19], we believe that the conformational changes observed in our study are likely broadly similar at forces below 5 pN.

Our study also revealed that magnesium is essential for ssDNA cleavage by both TRR and EcTopoI, in agreement with findings

from a previous study of EcTopoI using magnetic tweezers[18]. This contrasts with results from bulk biochemical experiments that suggested that magnesium affects only religation, and not the cleavage step[44,45]. The reason for this difference between the findings from single-molecule experiments and ensemble studies remains to be fully elucidated. On the other hand, our observation that magnesium does not influence the conformational changes associated with gate opening is consistent with previous studies of type 1A topoisomerases, that show that the key $Mg^{2+}$-interaction site (in the TOPRIM domain)[18,45–48] is located quite far from the primary hinge region (Fig. 7a) that governs gate opening[49].

The second feature of our approach is the use of fluorescence imaging to determine the successful binding of T-DNA to the topoisomerase core. In this way, we have revealed that both dsT-DNA and ssT-DNA can be successfully catenated by TRR-ssDNA. The fact that TRR can mediate the transfer of dsDNA through the cleaved ssDNA is particularly notable for the following reasons. First, the reported crystal structure for HsTopoIIIα-RMI1 shows that the decatenation loop associated with RMI1 protrudes into the central cavity of HsTopoIIIα such that the cavity has a maximum width of 11 Å[25]. On the basis of this, it has been suggested that steric constraints would disfavour dsT-DNA binding inside the TRR cavity[25]. Our results indicate, however, that there is sufficient plasticity in the TRR complex to accommodate dsT-DNA within the cavity, even with the gate closed—a necessary step for (de)catenation. This is consistent with previous bulk biochemical studies of ScTopoIII-Rmi1 which have indicated that dsT-DNA can be transferred through a TopoIII-mediated gate, albeit with reduced efficiency compared to ssT-DNA[20]. Furthermore, our findings, as well as previous experiments conducted with EcTopoIII[11], have shown that dsDNA can bind to the central cavity of the topoisomerase with high affinity. Since we observed a lower affinity of dsT-DNA for EcTopoI-ssDNA compared to TRR-ssDNA, we propose that dsT-DNA binding could be specific to TopoIII-like topoisomerases (e.g., EcTopoIII, ScTopoIII-Rmi1 and TRR) and may be relevant in vivo. A potential role of dsDNA transfer in vivo has recently been reported for EcTopoIII in resolving pre-catenanes originating during replication[50]. Pre-catenanes behind the replication fork contain ssDNA regions arising from Okazaki fragments on the lagging strand, that are in close proximity to dsDNA on the leading strand (Fig. 7b). These ssDNA regions could provide a binding target for EcTopoIII in order to transfer dsDNA[50]. Based on our results, we hypothesize that TopoIII-like topoisomerases, such as TRR, might play a similar role during replication in eukaryotic cells. Furthermore, dsDNA transfer by TRR could also be relevant for the resolution of late replication and/or recombination intermediates in vivo, including double Holliday junctions and hemi-catenanes, which also exhibit local regions of ssDNA in close proximity to dsDNA.

By comparing TRR and EcTopoI, we have revealed important similarities and differences between these type 1A topoisomerases. For example, while we found that most TRR-ssDNA gates are open at forces above at least 5 pN, the majority of EcTopoI-ssDNA gates are closed below 10 pN, in agreement with recent magnetic tweezers studies[19]. Our observation that most TRR-ssDNA gates are open at low forces is consistent with earlier structural predictions that the presence of the RMI1 co-factor (which is absent in EcTopoI/III) might stabilize gate opening[25]. Given that TRR is known to resolve ultra-fine anaphase bridges[28,51,52], which may be under tension[28,53,54], it is perhaps surprising that the enzymatic equilibrium appears to favour gate opening already at low tensions (from at least 5 pN). Nonetheless, transient gate closing is still likely to occur even at higher tensions, and this could potentially be regulated by other proteins.

In addition, TRR and its homologues are involved in several other processes in vivo, such as double Holliday junction dissolution[22,26,32,55,56], where the DNA tension is expected to be very low (less than a few pN).

Notably, we revealed an increase in the open gate size between 15 and 30 pN, which we refer to as gate widening, for both TRR-ssDNA and *Ec*TopoI-ssDNA. While it remains to be determined whether such gate widening plays an active role in vivo, it offers insight into the structural flexibility of the enzyme[6]. Moreover, the gate widening transition in *Ec*TopoI-ssDNA is associated with a larger length increase than for TRR-ssDNA. Given the strong sequence similarity between the cores of both enzymes[3,25], we hypothesize that the presence of RMI1 in TRR might reduce the flexibility of this topoisomerase, leading to less extensive gate widening. This is consistent with the fact that RMI1 binds close to a region of *Hs*TopoIIIα that is expected to serve as the pivot point responsible for gate opening[25]. Nevertheless, the gate widening transition might alternatively reflect the presence of a second hinge in the topoisomerase core structure[3,5,6,49].

In contrast to dsT-DNA binding, which increases the TRR-ssDNA gate size but otherwise does not change the overall gate opening mechanics, the interaction of BLM with TRR-ssDNA substantially alters the flexibility of the TRR-ssDNA gate, such that a distinction between gate opening and gate widening is not discernible. These observations might suggest that BLM binds in close proximity to the hinge(s) in TRR that facilitate(s) gate opening. This is of particular relevance since there is no available crystal structure of BLM in complex with TRR. The apparent reduction in flexibility of TRR-ssDNA by BLM is supported by the observation that BLM interacts with both *Hs*TopoIIIα and RMI1[29,34]. Strikingly, the influence of BLM on the TRR-ssDNA gate is largely reversed by the binding of dsT-DNA. Moreover, although both BLM and dsT-DNA can bind simultaneously to TRR-ssDNA, their affinity is reduced if the other is already bound to the TRR-ssDNA. This leads us to postulate that the function of TRR in vivo might be dependent on the local concentration of dsDNA. Transfer of dsDNA has been reported to be less efficient than for ssDNA in the case of yeast TopoIIIα-Rmi1[20]. However, our dsT/ssT-competition assay results suggest that the efficiency of dsT-DNA transfer by TRR could still be biologically relevant, in particular under conditions of a high local concentration of dsDNA (such as in pre-catenanes).

Taken together, our single-molecule approach provides insights into the gate opening and strand-passage pathways of TRR and reveals conformational changes within the TRR-ssDNA gate as a function of both tension and binding partners. We have demonstrated that TRR exhibits an unexpected structural plasticity in which the TRR-ssDNA gate size can be modulated in opposite ways by dsT-DNA and BLM, respectively. Furthermore, we have directly observed that TRR can mediate the transfer of dsDNA, which might indicate that this mechanism could play a greater role in vivo than previously appreciated.

## Methods

**Proteins**. TRR (unlabelled and labelled with mCherry), TRR-Y337F-mCherry, and BLM-SNAP649 were expressed and purified using established protocols[28]. In brief, wild-type or catalytically dead (Y337F) *Hs*TopoIIIα was overexpressed in *E.coli* Rosetta 2 (DE3, pLysS, Novagen) cells and purified as a pre-formed complex (TRR). The mCherry coding sequence was cloned on the N-terminus of RMI2. BLM-SNAP was expressed in JEL1 yeast cells and after purification, the protein was labelled with SNAP-Surface® 649. *E.coli* TopoI was purchased from New England Biolabs (NEB).

**Buffers and sample preparation**. Unless stated otherwise, all chemicals were purchased from Sigma Aldrich, while all enzymes and plasmids were purchased from NEB. λ-DNA (~48.5 kb, Roche) and pKYB1 (~8.3 kb) were end-labelled with biotin, as described previously[37]. For dsT-DNA binding experiments, pBR322 plasmid (4,361 bp) was used. Linearization of this plasmid was performed using

EcoR1-HF. In all cases where circular dsDNA was used, the plasmid was relaxed by reacting with *E. coli* TopoI. For ssT-DNA binding experiments, M13mp18 ssDNA plasmid (7,249 nt) was used. Both dsT- and ssT-DNA were stained with a fluorescent cyanine intercalating dye (SYBR Gold, Thermo Fisher Scientific) which is spectrally distinct from both mCherry and SNAP649. The short (30 nt) oligos were purchased from Biolegio BV. The ssDNA oligo sequence (5′-TTG TCC AAC TTG CTG TCC AGG TCG CCG CCC-3′) was fluorescently labelled at the 5′ end with ATTO647N. The dsDNA oligo was generated by hybridizing the 30 nt ssDNA oligo with its complementary strand. Generation of tethered ssDNA was achieved by force-induced melting of tethered dsDNA in a 'melting' buffer, which contained 5 mM HEPES pH 7.5 and 0.02% Tween 20, similar to previous reports[37]. In most other cases, experiments were performed in a 'standard' measurement buffer of 20 mM HEPES pH 7.5, 100 mM NaCl, 2 mM MgCl₂, 1 mM DTT, 0.02% Tween 20 and 0.05% Casein. For experiments conducted in a 'high-salt' buffer, the NaCl concentration in the standard buffer was increased to 500 mM. For experiments performed in the absence of magnesium, the standard buffer (but without MgCl₂) was supplemented with 1 mM EDTA.

**Single-molecule set-up**. Experiments were performed in a 6-channel microfluidic flow-cell (LUMICKS B.V.) that was mounted on an automated XY-stage housed within a custom-made inverted microscope that combines dual-trap optical tweezers with wide-field fluorescence microscopy[28]. In brief, two orthogonally polarized optical traps were generated using a 1064 nm fibre laser (YLR-10-LP, 10 W, IPG Photonics) via a water-immersion microscope objective (Plan Apo 60X, NA 1.2, Nikon). Biotinylated dsDNA (λ-DNA or pKYB1) was tethered between two streptavidin-coated optically trapped microspheres (Spherotech Inc) in situ within the flow-cell. Single TRR gate opening events (Fig. 2d) were recorded using pKYB1-ssDNA and 1.75 μm beads. For all other experiments, λ-ssDNA and 4.6 μm beads were used. All experiments were performed at room temperature. The optical tweezers were controlled using a custom-written programme written in LabVIEW 2011[28]. Fluorescence from TRR-mCherry, BLM-SNAP649, and SYBR Gold intercalator dye was generated by excitation with a 561 nm (Cobolt Jive 25 mW CW), a 639 nm (Coherent Cube 50 mW CW) and a 491 nm (Cobolt Calypso 50 mW CW) laser, respectively. Fluorescence images were recorded using an EMCCD camera (iXon + 897E, Andor Technology). Fluorescence excitation lasers were controlled with an AOTF (Acousto-optical tunable filter, AA Opto Electronic).

**Experimental conditions**. The concentration of TRR, *Ec*TopoI, and BLM was 20 nM, unless stated otherwise. In order to generate different TRR coverages on ssDNA (Fig. 2b), we used 4 nM TRR together with varying the incubation time. For recording single TRR-ssDNA gate opening events (Fig. 2d), a concentration of ~1 nM TRR was used. Fluorescence snapshots were recorded in a buffer lacking fluorescently labelled proteins to avoid background fluorescence. The concentration of T-DNA was 3–10 ng/μl, unless stated otherwise (note that 3 ng/μl was sufficient to result in saturation binding). For short ss/dsT-DNA experiments, an oligo concentration of 1 nM was used. The concentration of SYBR Gold dye used to stain T-DNA (either dsT or ssT) was 5–100 nM, depending on the salt concentration. The content of each channel in the microfluidic flow-cell for different single-molecule experiments is listed in Supplementary Table 1.

**Incubation procedure for single-molecule experiments**. After generating ssDNA in melting buffer, the molecule was moved into the TRR channel at a tension of 15 pN. Due to the extensive lengthening of ssDNA upon TRR-binding (in standard buffer containing magnesium), this resulted in a rapid reduction of the force to below 5 pN. The tethered TRR-ssDNA molecule was then moved into a different channel, either to record fluorescence snapshots/FD-curves or to incubate in a second reagent. In order to minimize the probability of interactions between excess dsDNA molecules tethered to the beads and the tethered TRR-ssDNA substrate (see Supplementary Fig. 3b), a modest tension (5 pN) was applied to the substrate when moving into different channels. However, unless stated otherwise, when incubating TRR/*Ec*Topo1-ssDNA in different channels, the force was substantially lower (~1–2 pN), although complete slack was avoided, again to minimize interactions with excess dsDNA molecules tethered to the beads. In order to minimize blurring caused by DNA fluctuations, all fluorescence snapshots were taken at a tension of ~5–10 pN, unless stated otherwise. FD curves were recorded at 50 Hz rate and an average DNA pulling speed of 2–5 μm/s.

**Data analysis**. FD-curves were extracted using a custom-written LabVIEW programme and analyzed using OriginPro 2016. Fluorescence images were analyzed using ImageJ based on published codes (https://imagej.nih.gov/ij/).

**Determination of TRR-ssDNA open gate size**. Distance–time traces were recorded for ssDNA in the presence of a sufficiently low concentration of TRR (~1 nM), such that a step-wise increase in distance over time was obtained. Discrete steps in these traces were identified using a step-fitting algorithm (custom-written in MATLAB R2015b, The MathWorks Inc) based on the change-point method[57–59]. Gate opening was assumed to correspond to breakpoints between segments. The threshold for a discrete step was set as four times the standard

deviation of the baseline signal (measured in the absence of protein), and discrete segments were defined as having a minimal duration of 2 s and a minimal change in distance of 0.003 μm.

**Subtraction data analysis**. In order to generate a subtraction plot, the relevant FD-curves were first inverted, yielding corresponding DF-curves. The difference in distance at a given force was then calculated and normalized relative to the contour length of bare λ-ssDNA (~27.5 μm).

**Reporting summary**. Further information on research design is available in the Nature Research Reporting Summary linked to this article.

## Data availability

The data that support this study are available from the corresponding authors upon reasonable request. All relevant data required to reproduce this work are in the main manuscript and Supplementary Information (Supplementary Figs. 1–7 and Supplementary Data 1–5). Source data are provided with this paper.

## Code availability

The MATLAB code for the step fitting algorithm that was used to fit the single gate opening events (inset Fig. 2d) is provided in the Supplementary Information (Supplementary Software 1).

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

## Acknowledgements

We are grateful to Daan Vorselen for generating the step-fitting algorithm, Dian Spakman for helpful feedback, and Giulia Bergamaschi for help with the figures. This work was supported by a Chemical Sciences Top grant from the Netherlands Organization for Scientific Research (NWO) (714.015.002, G.J.L.W, E.J.G.P, G.A.K), an NWO ENW grant (OCENW.GROOT.2019.012, G.J.L.W.), the NovoNordisk Foundation (Chromocapture; NNF18OC0034948: G.J.L.W., K.S., A.H.B., and I.D.H) and the Danish National Research Foundation (DNRF115) and Nordea Foundation (K.S., A.H.B., and I.D.H.).

## Author contributions

J.A.M.B., A.S.B., and G.A.K. designed and performed experiments, analyzed the data, and wrote the paper. P.C. performed experiments and data analysis. K.S. and A.H.B. purified proteins and edited the paper. I.D.H. initiated the project and edited the paper. G.J.L.W. and E.J.G.P. initiated the project, designed experiments, and wrote the paper. All authors discussed the results and commented on the manuscript.

## Competing interests

The combined optical tweezers and fluorescence technologies used in this article are patented and licensed to LUMICKS B.V., in which E.J.G.P. and G.J.L.W. have a financial interest. The remaining authors declare no competing interests.
