## [Peer Review File · Nature Communications]

Duplex DNA and BLM regulate gate opening by the human TopoIII α -RMI1-RMI2 complexReviewers' Comments:

Reviewer #1:

Remarks to the Author:

Bakx and colleagues describe single molecule experiments to study the mechanism of the human TopoIIIalpha/RMI1/RMI2 (TRR) complex. The TRR works together with the Bloom helicase (BLM) to resolve recombination and replication intermediates. The experiments are based on a combined instrument that allows visualization of single DNA molecules while maintaining them stretched using a pair of optical tweezers. In addition, use of several different adjacent microfluidic channels allows moving the molecule to different conditions during the experiment. The experimental setup is quite unique and allows the authors to test several different hypothesis about the mechanism of the TRR complex. The main results are in good agreement with what we expect for type IA topoisomerases and the TRR complex, but there are some interesting new findings. Overall, this is a very strong manuscript describing interesting and relevant work. Nevertheless, there are several point that need to be addressed.

1. The authors interpret the lengthening of the DNA upon TRR binding as due to the opening of the protein gate and leading to a 4 nm opening. I agree with this interpretation of the observations, but the authors should emphasize that this opening may only be happening as the DNA is stretched by a considerable force, at least 20 pN. It is not clear to me that under no force or a smaller force the gate would open in the same manner or to the same extent. Many other single molecule experiments on topoisomerases have been interpreted in the same manner, but I am still skeptical that the opening is in the manner depicted in the cartoons. The force experiments do suggest a maximum opening of the protein gate, but do they suggest that such a large opening is functionally relevant? Could the protein be moving in a different manner in the absence of a pulling force? These are issues worth discussing.
2. The experiments are always conducted in the presence of magnesium. The role of magnesium in the reaction is not completely clear and the authors have an opportunity to clarify it. If magnesium is solely associated with religation, length extension should be visible in the absence of magnesium. On the other hand, if magnesium is involved in cleavage, experiments with no magnesium should show no lengthening. Gunn et al. observed dissociation from DNA in the absence of magnesium, suggesting that magnesium is important for binding or cleavage. This is an important point that the authors could address easily.
3. The authors observe two peaks in the gate opening distance distribution. They attributed the second one to the opening by two TRR complexes. It is not clear what they mean by that. They need to clarify this explanation, maybe even provide a small supplemental cartoon figure.
4. The experiments showing catenation of ssDNA and dsDNA molecules are very important and support the functional role of TRR. The interpretation that the molecules are catenated is based on their inability to flow away from the stretched ssDNA molecule. The experiments would be a lot more convincing if the dsDNA circle was linearized and allowed to flow away, showing convincingly that it was indeed catenated. Cleaving the dsDNA molecules after catenation with a restriction enzyme at a single site and observing whether they flow away would strengthen the interpretation.
5. In Supplementary Note 3 the authors suggest the average area occupied as 450 nm. Do they mean area or length?
6. In Supplementary Figure 3 the upper and lower figures in panel d may be swapped.

Reviewer #2:

Remarks to the Author:

This paper summarizes combined force spectroscopy and fluorescence experiments on assemblies of human topoisomerase IIIalpha with RMI1 and RMI2 (TRR), bound to a template of single stranded DNA (ssDNA). As this looped protein binds, the ssDNA template is cleaved. As both ends of the open strand are held, the protein loop (the 'gate') opens, facilitating the entry and binding of another DNA strand. Both positive and negative DNA supercoiling are regulated by this important mechanism. The authors also repeat these measurements on E. coli TopoI, another type 1A topoisomerase.

The authors are able to quantitatively characterize the TRR gate opening length on ssDNA for the first time and compare this result with prior results for E. coli TopoI and TopoIII. They also directly observe for the first time the catenation of dsDNA and ssDNA by TRR. The dramatic effect of dsDNA on this behavior is surprising because crystal structures suggested that this topoisomerase would not be able to accommodate dsDNA. They also show that another factor BLM modulates gate opening. This is an intriguing set of results that will certainly be of interest to the readers of this journal. The experiments are thoughtfully presented and carefully explained, and the fluorescence and force spectroscopy data are skillfully combined. With some specific clarifications to the results and discussion, we recommend publication.

Specific Comments:

Abstract, Page 3 and Page 6: The measured and reported values of the open gate size should be stated more clearly here, especially in the results section. For TRR, the open gate is found in this work to be 4.0 ± 1.5 nm. In ref 19, the values were found for EcTopoI and EcTopoIII separately, with uncertainty, and these numbers should be specifically shown in the results for the most meaningful comparison. Note: in ref 19 the E coli TopoI and III use roman numerals, but here the authors use 1 and 3 for the prokaryotic enzymes. What is the standard for this notation?

Page 5 and Figure 2: The fluorescence images in panel A are all taken at 10 pN, according to the methods. Was the fraction bound independent over the force ranges shown here? Could 'low', 'medium' and 'high' be specified?

Page 6 and Figure 2: If the shift shown in panel B is a function of the fraction bound, then is the shift shown in A at saturation, corresponding to the 'high' image?

Page 6 and Figure 2: The caption to panel C reports open gate size with uncertainty derived from the standard deviation, while elsewhere in the manuscript the error in the mean is used. Clearly the error in the measured average is known to better uncertainty. Furthermore, don't the two peaks imply two results, which would be averaged with uncertainty? It may be important to point out the SD to show the distribution of steps, but when comparing with the E coli results SEM may be more appropriate.

Page 6 and Figure 2: Could the authors share more detailed information on the step fitting algorithm of panel C, even if in the supplement?

Page 6 and page 7-8: The previous study noted above showed not only much larger opening gate sizes but found that while most EcTopo I binding events led to opening, only a fraction of bound EcTopo III did. Was that the case here? A later result showed a further length increase upon dsDNA binding, which was attributed to additional gate opening, as opposed to an increased fraction of opening/binding. While a compelling case is made for additional opening, this discrepancy should be addressed in the main text.

Page 8 and Figure 4: In panel A the measured end to end distance was confusing. This number should be force dependent. Is it a change in contour length from a fit? What is the uncertainty?

Reviewer #3:

Remarks to the Author:

Bakx and coworkers used a combination of optical trapping and fluorescence to study the binding and gate opening of the human topoisomerase III -RMI1-RMI2 (TRR) complex to ssDNA held between two optically trapped beads. They observe an increase in the extension of the ssDNA in the presence of

TTR that they interpret as gate opening. They then study the binding of long circular ssDNA and dsDNA molecules to TRR molecules in an open gate conformation, and demonstrate religation of TRR through trapping of the circular molecules on the ssDNA template. Finally, they compare the gate opening distance of TRR to E coli topoisomerase I and the full Bloom's helicase, TRR (BTTR) complex in the presence of ds and ss DNA molecules.

The main results constitute the observation of the gate opening distance of the TRR complex, that this distance is not substantially increased by binding of ssDNA to the open gate, but is slightly increased by binding of dsDNA to the open gate, and that the gate opening is slightly smaller in the BTTR complex. The conclusion is that the extent of gate opening is somewhat plastic and that this may be important in vivo.

Whereas the technique and technical aspects of the measurements are, with a few exceptions, of the very highest caliber, the work suffers from errors in logic and over-interpretation of the results. As described in detail below, the results although interesting do not probe the catalytic cycle of the enzyme and hence the connection to the biology and function of the TRR and BTTR complexes are tenuous at best. The system described has enormous potential to provide mechanistic details of the TRR and BTTR gate opening and strand passage catalytic cycle, but the current data and measurements do not speak to the biology of the system.

Main points

1. The authors claim that the binding of dsDNA and Bloom's helicase alters the extent of gate opening, but this is probed in the state after the gate is open – the crucial step in strand passage and the opening of the gate takes place before the transfer strand enters the cavity and can bind to the topoisomerase. They are probing the very last step in the catalytic cycle in which the important steps precede the step they are probing. The fact that the gate is locked in the open state at the forces at which the experiments are conducted means that they are simply probing the effects on binding at the end of the catalytic cycle when the crucial cleavage and strand passage reactions have already taken place, rather than the extent of gate opening during strand passage. Another issue with the data in relation to the biological strand passage reaction is that all the relevant measurements have been made at a force at which the gate is locked open. By construction this cannot represent a physiological condition, since the gate can never close and the catalytic cycle can never be completed. As a result, the important biological processes are occurring precisely at a regime where the measurements cannot probe the gate opening extent or kinetics. For these reasons the measurements are probing completely unphysical aspects of the gate opening process. The data do not report on what is happening in the regime at which the TRR or BTTR complex can actually complete the strand passage reaction and so unfortunately the data are difficult to interpret or relate to what is actually happening in the actual biological process. This is particularly problematic for the case of BTTR where in figure 6 B it appears that the gate opening is in fact larger at low forces, closer to where the enzyme may in fact operate, rather than smaller as is observed at higher forces, which are well beyond the range of forces where the enzyme could operate.
2. There is a lot of speculation about the impact of the observations on the decatenation activity of the TRR or BTTR complex, but these are not tested through biochemical experiments. Given the issues raised above, the conclusions from the single-molecule measurements are problematic, but these conclusions would nevertheless be relatively straightforward to test directly.
3. The results raise some interesting and perhaps troubling questions concerning the putative roles of the TRR and BTTR complex in resolving anaphase bridges or other DNA links that are under elevated levels of stress or tension. The results suggest that the topoisomerase III gate cannot close against a force of ~ 5 pN or perhaps less. The authors fail to address this seeming contradiction between their data and the physiological roles of this complex. The results in this respect are surprising and intriguing, but this aspect is largely ignored.
4. There are a number of technical and interpretation issues that are described in detail below but which together suggest that the interpretation of the data is somewhat superficial and the findings are less clear-cut and straightforward than they are presented.

Detailed points:

Line 293: it is not entirely clear how this technique probes the steps of the catalytic cycle. One typically associates the catalytic cycle with the kinetic steps of the cycle. This assay probes the

conformational change associated with gate opening and monitors a single round of catenation mediated by the presumably locked open gate. The assay therefore probes the very last step of the process and the changes in gate opening correspond to the final step of the process, just prior to gate closure, religation and release.

Line 297: This technique does not measure kinetics. This is an inaccurate statement and deters from the potential impact and significance of this work.

Line 301: The measurement of the change in gate size with dsT-DNA binding is inaccurate unless individual changes in extension were measured since it is highly unlikely that every bound topoisomerase also bound a dsT-DNA segment. I would argue that the authors have demonstrated that dsT-DNA binding induces an increase in the gate opening, but the measurement represents a lower bound on the actual increase in opening given the uncertainty in the number of bound topoisomerases that also bound dsT-DNA. The increase in opening is an interesting measurement but the caveats of the extent of this opening need to be more fully discussed. This point is addressed in the supplemental information but should be included in the main text. The authors correctly state that the increase in extension is an lower bound on the increase.

Figure 2: Based on the nominal increase in extension per bound topoisomerase at low forces, can the authors give some sense of the scale and probability of the additional opening transition above 20 pN. Is this gradual opening a kinetic effect or is it a continuous force dependent spring like extension that kicks in above 20 pN? From the curve it appears that there is a second kink in the curve around 30 pN – does the curve above ~ 30 pN map back onto the ss-DNA curve?

Figure 2B – at what force was this extension vs fluorescence intensity measured? What range of topoisomerase 3 concentration was used for these measurements? What are the error bars on the measurements of both elongation and fluorescence intensity?

Figure 2C- what are the mean values (+/- uncertainties) of the two Gaussians?

Figure 3: What concentration of ds and ssDNA were used in these example measurements and what concentration of TRR was used? Without titration or other data, the change in extension associated with ssDNA or dsDNA binding are strictly lower bounds. There is a clear increase in extension for dsDNA, but unless this extension is shown to be independent of the concentration of dsDNA then the increase in extension represents an unknown number of TRR molecules that have bound dsDNA. The authors correctly note this issue in the supplemental information (Supp note 4), but this should be more clearly stated in the main text.

Figure 3B. The dsDNA bound DNA curve appears to have three distinct regimes – there appears to be a kink in the curve around ~ 9 pN and another kink in the curve around 30 pN.

Figure 4A. Can the authors comment on the fact that the TRR curve after the 500 mM NaCl wash overlaps the ssDNA curve to a significant extent at forces up to about 10 pN? This seems to be at odds with the observation that the TRR curve shifted to the right by a fixed amount as was the case for the initial TRR curve in green, and the similar curves in Figures 2 and 3.

The ss and ds T-DNA binding experiments to the open gate are the most novel and unique aspects of this work. The authors have the unique ability to probe this interaction that is typically transient and impossible to capture. Can their data be interpreted in relation to the relative affinity of the enzyme cavity for the different DNA substrates? It is also possible that the affinity of the open enzyme for ssDNA or dsDNA is different – and the authors have the possibility to probe these differences. This would probe a unique and difficult to measure aspect of the process and could indirectly shed light on the strand transfer process. These measurements would be highly meaningful and informative.

Figure 5F. The caption states that “ λ -ssDNA (stained with intercalator dye)” but I think that the authors meant that the circular ssT-DNA was stained.

Figure 6B. Once again the TRR curve shown in this FD curve appears to behave very differently than the ssDNA curve. There appears to be a significant kink in the curve at ~ 9 -10 pN followed by a convex curvature to ~ 25 pN. It seems that TRR behaves differently in each of these measurements and does not consistently simply shift the ssDNA FD curve to the right as stated in Figure 2.

Furthermore, the addition of Bloom's helicase induces changes in extension that are substantially more subtle than simply decreasing the extension. Indeed, at low force BLM seems to increase the extension, and then depending on the applied force, there seems to be a variable change in the

relative extension between the red and green curves. These changes in relative extension indicate that the overall response to force has changed. This is not simply a change in the average gate opening but a complete reshaping of the force-dependent opening of the complex. The most striking observation is that at low force, the most relevant *in vivo* since it is closer to the force at which the gate can presumably close, the extension has increased with the addition of BLM. The complexity of the BTTR vs the TRR curve begs the question as to how the change in extension between the two complexes was determined. Is this an average difference between the two curves or an extreme value, as represented by the red arrow? Given this curve, it seems to be a misstatement to characterize the difference between TRR and BTTR as simply an decrease in gate opening extent – particularly since the opposite appears to be the case at lower forces where the enzyme could actually close the gate. It would be illustrative to see the difference in extension plotted as a function of force. Along the same lines, the claim in figure 2 is that the TRR curve overlaps that of ssDNA but with an offset, if this is generally true then it would be instructive to fit the curves and report on this offset and also the degree to which the TRR and BTTR curves are indeed represented by an off-set ssDNA curve rather than something more complex.

Supp figure 1. The abbreviations (AOTF, CMOS, EIS, etc) should be spelled out in the caption.

Supp Figure 2. There are no kinetic measurements in this figure so the title is misleading. At best panels d and e support the claim that the opening measurements are at equilibrium, but there are no actual kinetic measurements. In panel a there is a small but reproducible downward curvature in the FD curves between ~ 10 and 24 pN, which is not consistent with a ssDNA stretching curve, this is particularly clear in panel d, but less pronounced in panel e. Together these suggest that there are internal motions or transitions that are occurring as a function of force that represent a more complex process than a simple gate opening.

Supp figure 3. If I understand panel B correctly, then there is likely an excess of dsDNA on the two optically trapped beads that may interact with TRR. In this image these interactions are visualized but it seems likely or possible that this occurs in all of the experiments since the DNA substrate begins as a double stranded DNA and is mechanically manipulated to create a ssDNA. In this case, it seems that both beads could be bound to excess dsDNA molecules that could interact with TRR or BTTR or topo I and alter the reported results. In panel e both fits are labeled “y” in principle one of them should be “x”.

Supp Figure 6 a. Given the differences between the TRR and BTTR curves, how was the difference in length measured? At what force was the difference calculated, or was the difference in length averaged over different forces? Please indicate how the difference in length was obtained from the FD curves that have different shapes and different distances between them, cf panel d.

In sum, despite the overall high quality of the data and the results, I am not convinced that the work sheds light on the catalytic cycle of topoisomerase III under physiological conditions and therefore fails to contribute significantly to, or meaningfully advance, the field.

Reviewer #1:

Bakx and colleagues describe single molecule experiments to study the mechanism of the human TopoIIIalpha/RMI1/RMI2 (TRR) complex. The TRR works together with the Bloom helicase (BLM) to resolve recombination and replication intermediates. The experiments are based on a combined instrument that allows visualization of single DNA molecules while maintaining them stretched using a pair of optical tweezers. In addition, use of several different adjacent microfluidic channels allows moving the molecule to different conditions during the experiment. The experimental setup is quite unique and allows the authors to test several different hypothesis about the mechanism of the TRR complex. The main results are in good agreement with what we expect for type IA topoisomerases and the TRR complex, but there are some interesting new findings. Overall, this is a very strong manuscript describing interesting and relevant work. Nevertheless, there are several point that need to be addressed.

1. The authors interpret the lengthening of the DNA upon TRR binding as due to the opening of the protein gate and leading to a 4 nm opening. I agree with this interpretation of the observations, but the authors should emphasize that this opening may only be happening as the DNA is stretched by a considerable force, at least 20 pN. It is not clear to me that under no force or a smaller force the gate would open in the same manner or to the same extent. Many other single molecule experiments on topoisomerases have been interpreted in the same manner, but I am still skeptical that the opening is in the manner depicted in the cartoons. The force experiments do suggest a maximum opening of the protein gate, but do they suggest that such a large opening is functionally relevant? Could the protein be moving in a different manner in the absence of a pulling force? These are issues worth discussing.

We are grateful for the reviewer's helpful feedback. In light of these comments, we realize that we could have explained more clearly that the observed gate opening does not require application of significant force. While our single-molecule determination of the open-gate size was performed at 15 pN, the increase in ssDNA length due to TRR binding is roughly constant from low forces up to ~15 pN. Nevertheless, we appreciate that this may not have been sufficiently clear. To demonstrate this more directly (and also as a response to points raised by the other reviewers), we have now plotted the difference in length between TRR-ssDNA and bare ssDNA as a function of force (based on the corresponding FD-curves). These so-called 'subtraction plots' demonstrate more clearly that the length shift due to TRR binding is constant from at least 5 pN (which is as low as we can reliably measure) up to ~15 pN. We have included this subtraction plot in our revised version of Fig. 2 (panel c). We acknowledge that we cannot comment with certainty about the conformational state of the protein at forces below 5 pN (as discussed in our new Supplementary Note 3). Thus, it is possible that the mechanics of gate opening could be different at forces below 5 pN. Nevertheless, we suggest that this is unlikely for the following reasons: (i) 5 pN is still a relatively low force, in terms of both the forces encountered *in vivo* and the forces typically required to induce non-physiological conformational changes of proteins; (ii) there is no indication that the gate size is force-dependent until a critical force of ~15 pN, suggesting that the open state has a stable and well-defined structure; and (iii) molecular simulations by Mills *et al.*, NSMB 2018 (for *Ec*TopoI and *Ec*TopoIII) suggest that substantial gate opening is required to pass a T-DNA strand through the gate.

*2. The experiments are always conducted in the presence of magnesium. The role of magnesium in the reaction is not completely clear and the authors have an opportunity to clarify it. If magnesium is solely associated with religation, length extension should be visible in the absence of magnesium. On the other hand, if magnesium is involved in cleavage, experiments with no magnesium should show no lengthening. Gunn *et al.* observed dissociation from DNA in the absence of magnesium, suggesting that magnesium is important for binding or cleavage. This is an important point that the authors could address easily.*

We thank the reviewer for this helpful suggestion. Based on the reviewer's comments, we have performed a series of new experiments in which we recorded FD-curves of TRR-ssDNA in the presence and absence of magnesium. Here, two distinct experiments were performed. In the first, a tethered ssDNA molecule was incubated in a buffer containing TRR with no magnesium (supplemented with EDTA) and then moved (at low force) to a buffer containing magnesium. The

FD-curve recorded in the TRR channel in the absence of magnesium showed no length increase, despite confirmation of successful protein binding using fluorescence microscopy. However, once the TRR-coated ssDNA molecule was moved to the magnesium-containing buffer, the corresponding FD-curve displayed the characteristic lengthening associated with TRR-ssDNA gate opening. This indicates that the presence of EDTA inhibits either ssDNA cleavage and/or gate opening. To differentiate between these two possibilities, we performed a second experiment, in which a tethered ssDNA molecule was incubated in a buffer containing both TRR and magnesium and then moved to a buffer lacking magnesium (supplemented with EDTA). The FD-curves recorded in both buffers displayed a substantial increase in length (which we ascribe to gate opening), suggesting that magnesium is crucial for ssDNA cleavage, but not for gate opening. The results from these experiments are shown in our revised version of Fig. 2 (panels e and f) and described in the revised Results section (*'ssDNA cleavage, but not gate opening, requires magnesium'*). We also performed similar experiments for *Ec*TopoI and obtained similar findings (reported in Supplementary Fig. 6), suggesting that this behaviour is a common feature of type 1A topoisomerases.

3. The authors observe two peaks in the gate opening distance distribution. They attributed the second one to the opening by two TRR complexes. It is not clear what they mean by that. They need to clarify this explanation, maybe even provide a small supplemental cartoon figure.

In our original manuscript, the gate opening distribution displayed two peaks. The second peak was ascribed to gate opening by two TRR complexes simultaneously bound at two different sites on the DNA. Such a double Gaussian distribution was thought to be due to the fact that, although the TRR concentration was very low, there is still a chance that two TRR complexes could bind to the ssDNA molecule (each inducing gate opening events) in such quick succession that their rate of binding and gate opening exceeds the time resolution of our instrument. However, during the revision of our manuscript, we critically re-evaluated our step-fitting analysis approach and concluded that the experimental noise in our instrument was higher than initially estimated. We have now re-assessed our criteria for the step-fitting algorithm (which is now detailed in our revised Methods section) by setting a higher minimum step-fitting threshold. This new analysis indicates that, rather than having two peaks, the gate opening distribution is in fact most likely a single broad Gaussian distribution, with a peak at $\sim 8.5 \mu\text{m}$. We have revised Fig. 2 and the accompanying text (in the both the Results and Methods sections) to reflect this. Although the value of the open gate size changes substantially in the revised version of our manuscript, this does not alter the conclusions of our study. This is because we have focussed our study on another, much more robust, approach to quantify gate opening, namely by analyzing FD-curves. In the latter approach, we determine the average behaviour of over 1000 proteins, and this method is thus much less sensitive to instrument noise compared to measuring single gate opening events and fitting the corresponding distribution. In summary, our reassessment of the open gate size has no influence on the rest of our conclusions (for example, the gate size does not feature in the Discussion section of either the original or the revised manuscript).

4. The experiments showing catenation of ssDNA and dsDNA molecules are very important and support the functional role of TRR. The interpretation that the molecules are catenated is based on their inability to flow away from the stretched ssDNA molecule. The experiments would be a lot more convincing if the dsDNA circle was linearized and allowed to flow away, showing convincingly that it was indeed catenated. Cleaving the dsDNA molecules after catenation with a restriction enzyme at a single site and observing whether they flow away would strengthen the interpretation.

We agree that comparing the catenation efficiency of circular and linear dsDNA would be informative and we thank the reviewer for this helpful feedback. The reviewer's suggestion to catenate circular dsDNA and subsequently linearize this using a restriction enzyme is, unfortunately, too complex to conduct in our current experimental set-up. Nevertheless, as an alternative, we have repeated the experiments reported in Fig. 4, using linear, rather than circular, dsDNA and show this in Fig. 4e of our revised manuscript. In this way, we reveal two important findings: (i) the fraction of dsDNA remaining after incubation of the TRR-ssDNA in high salt is much lower when using linear, rather than circular, dsDNA and (ii) the small fraction of linear dsDNA that remains on the ssDNA is immobile. The latter observation is inconsistent with catenated DNA (which we have shown can

diffuse rapidly over the ssDNA molecule). Rather, it suggests that the remaining linear dsDNA molecules are bound to residual TRR complexes on the ssDNA and are not catenated to the ssDNA. Together, these findings support the notion that the circular dsDNA molecules are catenated with the ssDNA.

5. In Supplementary Note 3 the authors suggest the average area occupied as 450 nm. Do they mean area or length?

We thank the reviewer for bringing this to our attention. We were referring to length, rather than area, and have clarified this in the revised Supplementary Note 5 (Supplementary Note 3 in our original manuscript).

6. In Supplementary Figure 3 the upper and lower figures in panel d may be swapped.

We are grateful to the reviewer for highlighting that the caption and figure for Supplementary Fig. 3d were not consistent. We have adjusted the caption to correct for this.

Reviewer #2:

This paper summarizes combined force spectroscopy and fluorescence experiments on assemblies of human topoisomerase IIIalpha with RMI1 and RMI2 (TRR), bound to a template of single stranded DNA (ssDNA). As this looped protein binds, the ssDNA template is cleaved. As both ends of the open strand are held, the protein loop (the 'gate') opens, facilitating the entry and binding of another DNA strand. Both positive and negative DNA supercoiling are regulated by this important mechanism. The authors also repeat these measurements on E. coli TopoI, another type IA topoisomerase.

The authors are able to quantitatively characterize the TRR gate opening length on ssDNA for the first time and compare this result with prior results for E. coli TopoI and TopoIII. They also directly observe for the first time the catenation of dsDNA and ssDNA by TRR. The dramatic effect of dsDNA on this behavior is surprising because crystal structures suggested that this topoisomerase would not be able to accommodate dsDNA. They also show that another factor BLM modulates gate opening. This is an intriguing set of results that will certainly be of interest to the readers of this journal. The experiments are thoughtfully presented and carefully explained, and the fluorescence and force spectroscopy data are skillfully combined. With some specific clarifications to the results and discussion, we recommend publication.

Specific Comments:

Abstract, Page 3 and Page 6: The measured and reported values of the open gate size should be stated more clearly here, especially in the results section. For TRR, the open gate is found in this work to be 4.0 +/- 1.5 nm. In ref 19, the values were found for EcTopoI and EcTopoIII separately, with uncertainty, and these numbers should be specifically shown in the results for the most meaningful comparison.

We fully agree with the reviewer and have revised our manuscript to state the uncertainty in both our determination of the open gate size of TRR and of the open gate size reported previously for EcTopoI and EcTopoIII.

Note: in ref 19 the E coli TopoI and III use roman numerals, but here the authors use 1 and 3 for the prokaryotic enzymes. What is the standard for this notation?

To the best of our knowledge there is no standard notation. We previously chose to use Arabic numerals for E coli enzymes and Roman numerals for eukaryotic enzymes, partly to distinguish between prokaryotic and eukaryotic systems and also because this is consistent with most (but not all) of the literature cited in our study. However, we note that in the wider literature, both Roman and Arabic numerals are used interchangeably to describe type IA topoisomerases. In light of the reviewer's question, we feel that it would be preferable to be consistent in our notation throughout our article, by naming both E coli and eukaryotic enzymes with Roman numerals. We have made this change in our revised draft, but added a statement in the introduction to acknowledge that E coli TopoI/III is often also referred to as Topo1/3.

Page 5 and Figure 2: The fluorescence images in panel A are all taken at 10 pN, according to the methods. Was the fraction bound independent over the force ranges shown here? Could 'low', 'medium' and 'high' be specified?

We thank the reviewer for highlighting these issues. The fluorescence images in Fig. 2a of the original manuscript were obtained at ~10 pN (actually ~5-10 pN, as is now stated in the revised Methods section) in order to minimize blurring due to Brownian fluctuations that can occur at lower forces. However, this is not the force at which the ssDNA was incubated in the protein channel (we explain this now more clearly in the revised Methods). When moving a bare ssDNA substrate into the TRR channel, the force was ~15 pN, in order to prevent DNA secondary structure formation (e.g. hairpins). However, upon TRR binding, the force reduced rapidly (<10 s) to below 5 pN. Using this approach, a concentration of 20 nM TRR was sufficient to achieve saturation (Supplementary Note 2). Hence, there is no indication that the use of higher forces would lead to increase binding. Nonetheless, when the ssDNA substrate was initially moved into the TRR channel at forces below 5 pN, TRR binding was reduced, presumably due to the presence of hairpins. We now mention this explicitly in Supplementary Note 8. Regarding our definitions of 'low', 'medium' and 'high' coverage, please see our answer to the following question directly below.

Page 6 and Figure 2: If the shift shown in panel B is a function of the fraction bound, then is the shift shown in A at saturation, corresponding to the 'high' image?

The three fluorescence snapshots that were originally depicted in Fig. 2 were not obtained from the same molecule as that used to record the FD-curve shown in panel a. Rather, they corresponded to the data used in panel b. The snapshot originally denoted as 'high' coverage had a fractional binding of ~22% (thus it would have been more reasonable to call this 'medium' coverage). However, in light of the reviewer's comments, we now instead show only a single fluorescence snapshot, the one that was obtained for the same molecule as that used to record the FD-curve in Fig. 2a. This snapshot corresponds to near-saturated TRR binding (on the basis that the length shift is close to the maximum we can detect). We believe that this change in data representation provides a more informative figure for the reader.

Page 6 and Figure 2: The caption to panel C is reports open gate size with uncertainty derived from the standard deviation, while elsewhere in the manuscript the error in the mean is used. Clearly the error in the measured average is known to better uncertainty. Furthermore, don't the two peaks imply two results, which would be averaged with uncertainty? It may be important to point out the SD to show the distribution of steps, but when comparing with the E coli results SEM may be more appropriate.

As outlined in our response to Reviewer 1, we have re-evaluated our step-fitting analysis approach and concluded that the experimental noise in our instrument was higher than initially estimated. Accordingly, we have now re-assessed our criteria for the step-fitting algorithm (which is now detailed in our revised Methods section) by setting a higher minimum step-fitting threshold. This new analysis indicates that, rather than having two peaks, the gate opening distribution is in fact most likely a single broad Gaussian distribution, with a peak at ~8.5 μm . With regards to our estimated error on this step size value, we prefer to use SD, rather than SEM, for two reasons. First, SD was used by Mills *et al.* (NSMB, 2018) when reporting the gate size of both *EcTopoI* and *EcTopoIII*. Second, our distribution of step sizes is quite broad and we cannot exclude the possibility that sub-steps or multiple steps could be contributing to this distribution. We feel that the (larger) error associated with the SD will reflect this uncertainty more appropriately.

Page 6 and Figure 2: Could the authors share more detailed information on the step fitting algorithm of panel C, even if in the supplement?

The step fitting algorithm and parameters are now described in detail in the revised Methods section.

Page 6 and page 7-8: The previous study noted above showed not only much larger opening gate sizes but found that while most EcTopo I binding events led to opening, only a fraction of bound EcTopo III did. Was that the case here? A later result showed a further length increase upon dsDNA binding, which was attributed to additional gate opening, as opposed to an increased fraction of

opening/binding. While a compelling case is made for additional opening, this discrepancy should be addressed in the main text.

The reviewer refers to the fraction of TRR-ssDNA gates that are open, and asks how this compares with published data for *Ec*TopoI. We note that in Fig. 2 from Mills *et al.* (NSMB, 2018), it was reported that ~90% of *Ec*TopoI-ssDNA gates are open, in contrast to *Ec*TopoIII-ssDNA, where only ~10% of gates are open. It is important to appreciate that these published values were acquired at a force of 22-24 pN. If we compare our TRR data at the same forces, we conclude that essentially all TRR-ssDNA gates are open, and substantial gate-widening has also occurred. At much lower forces (e.g. ~5 pN), both our data and those of Mills *et al.* indicate that most *Ec*TopoI-ssDNA gates are closed (as discussed in our new Supplementary Note 4). Thus, our results for *Ec*TopoI-ssDNA are in good agreement with those of Mills *et al.* Interestingly, our data reveal that most TRR-ssDNA gates – in contrast to both *Ec*TopoI-ssDNA and *Ec*TopoIII-ssDNA – are open even at low forces (from at least 5 pN). To demonstrate this difference between *Ec*TopoI and TRR more clearly, we now present subtraction plots (Fig. 5b), showing the difference in length between enzyme-bound ssDNA and bare ssDNA as a function of force (derived from the corresponding FD-curves). These highlight that TRR-ssDNA displays a constant length increase from at least 5 pN to 15 pN, whereas significant length increase is only observed for *Ec*TopoI-ssDNA at forces higher than ~10 pN. As we now highlight in the 3rd paragraph of our revised Discussion section, we attribute this significant difference in protein mechanics to the presence of the RMI1 co-factor in the case of TRR.

Page 8 and Figure 4: In panel A the measured end to end distance was confusing. This number should be force dependent. Is it a change in contour length from a fit? What is the uncertainty?

We thank the reviewer for highlighting this point and we now state the TRR-induced lengthening as a fractional percentage, rather than in micrometres. As explained above, the lengthening is independent of force from at least 5 pN to 15 pN. Regarding the error, the values stated in Fig. 4a are deduced from the corresponding curves and therefore can be determined with high precision (<<1%). There is, nonetheless, some variation in the fractional unbinding from molecule to molecule, but on average $71 \pm 6\%$ of TRR dissociates in high salt. We have now stated this in our revised Results section ('Direct visualization of ds/ssT-DNA catenation with ssDNA').

Reviewer #3:

Bakx and coworkers used a combination of optical trapping and fluorescence to study the binding and gate opening of the human topoisomerase III -RMI1-RMI2 (TRR) complex to ssDNA held between two optically trapped beads. They observe an increase in the extension of the ssDNA in the presence of TRR that they interpret as gate opening. They then study the binding of long circular ssDNA and dsDNA molecules to TRR molecules in an open gate conformation, and demonstrate religation of TRR through trapping of the circular molecules on the ssDNA template. Finally, they compare the gate opening distance of TRR to E coli topoisomerase I and the full Bloom's helicase, TRR (BTTR) complex in the presence of ds and ss DNA molecules.

The main results constitute the observation of the gate opening distance of the TRR complex, that this distance is not substantially increased by binding of ssDNA to the open gate, but is slightly increased by binding of dsDNA to the open gate, and that the gate opening is slightly smaller in the BTTR complex. The conclusion is that the extent of gate opening is somewhat plastic and that this may be important in vivo.

Whereas the technique and technical aspects of the measurements are, with a few exceptions, of the very highest caliber, the work suffers from errors in logic and over-interpretation of the results. As described in detail below, the results although interesting do not probe the catalytic cycle of the enzyme and hence the connection to the biology and function of the TRR and BTTR complexes are tenuous at best. The system described has enormous potential to provide mechanistic details of the TRR and BTTR gate opening and strand passage catalytic cycle, but the current data and measurements do not speak to the biology of the system.

Main points

1. The authors claim that the binding of dsDNA and Blooms helicase alters the extent of gate opening, but this is probed in the state after the gate is open – the crucial step in strand passage and the opening of the gate takes place before the transfer strand enters the cavity and can bind to the topoisomerase. They are probing the very last step in the catalytic cycle in which the important steps precede the step they are probing.

The reviewer suggests that we are probing only the ‘last step’ of the catalytic cycle and that this is the least important. We would first like to stress that we do not only probe the last step, but also provide valuable information about several essential steps in the catalytic cycle. For example, we demonstrate that a non-cutting mutant of TRR undergoes the first step of the cycle, *i.e.*, protein binding, but is not able to undergo the second step, *i.e.*, ssDNA-cleavage (Supplementary Fig. 2c). In our revised manuscript, we additionally reveal that magnesium plays a crucial role in ssDNA-cleavage by wild-type TRR, but has no detectable influence on gate opening (Fig. 2e, f). This experiment was a request from Reviewer 1 and addresses an important open question in the field. Additionally, our assay allows us to differentiate between T-DNA binding (Fig. 3) and catenation (Fig. 4). Moreover, co-factor-binding to the open TRR-ssDNA gate represents an important step in the catalytic cycle. Our study provides the first direct evidence that dsT-DNA can efficiently bind to the open gate and compete with ssT-DNA binding. This raises the possibility that dsDNA transfer by TRR is more relevant *in vivo* than had been appreciated previously. Using our technique, we are furthermore able to reveal major differences between the fractional gate opening of TRR and *EcTopo1*, and demonstrate that the latter is not able to efficiently catenate dsT-DNA (Fig. 5). Finally, the fact that both dsT-DNA and BLM (Fig. 6) alter the mechanical properties of the open TRR-ssDNA gate is a novel and important finding that provides insight into the structural plasticity of the gate that would be difficult to obtain with other techniques.

The fact that the gate is locked in the open state at the forces at which the experiments are conducted means that they are simply probing the effects on binding at the end of the catalytic cycle when the crucial cleavage and strand passage reactions have already taken place, rather than the extent of gate opening during strand passage.

The application of force can often alter the population between different protein states (in a way equivalent to altering temperature or concentrations), which allows us (and other people using force spectroscopy) to measure the properties of these states in more detail than could be obtained otherwise. We acknowledge the reviewer’s concern that the application of force, in the case of TRR-ssDNA, could potentially bias the gate opening equilibrium significantly. We believe, however, that this is not the case and that the conclusions we draw from our force measurements are also valid in the absence of force, for the following reasons. As addressed in a new Supplementary Note 4, we estimate that at least 38% of all TRR-ssDNA gates are open even in the absence of tension at any given time. This estimate is obtained by applying the force dependence previously established for *EcTopoI* (Mills *et al.*, NSMB, 2018) to TRR. Given that we anticipate that a gate opening event of even a short duration (~1 s) should be sufficient for successful strand transfer, we argue that our main conclusions are equally valid in the absence of force. One potential exception is the case of dsT-DNA, since that induces additional gate opening, and thus might be influenced by force. For this reason, we conducted a control experiment in which the TRR-ssDNA was incubated in dsT-DNA at forces below 2 pN. As shown in a revised version of Supplementary Fig. 4 (panel e) and discussed in Supplementary Note 8, this reveals that even under these conditions, there is still substantial dsT-DNA binding. Consequently, we do not believe our results and conclusions are significantly altered by the application of force.

Another issue with the data in relation to the biological strand passage reaction is that all the relevant measurements have been made at a force at which the gate is locked open. By construction this cannot represent a physiological condition, since the gate can never close and the catalytic cycle can never be completed. As a result, the important biological processes are occurring precisely at a regime where the measurements cannot probe the gate opening extent or kinetics. For these reasons the measurements are probing completely unphysical aspects of the gate opening process. The data do not report on what is happening in the regime at which the TRR or BTTR complex can actually

complete the strand passage reaction and so unfortunately the data are difficult to interpret or relate to what is actually happening in the actual biological process.

Here, the reviewer seems to have concerns about the biological relevance of our approach, in particular with respect to the forces we apply. As explained above, the assumption that we artificially force the gates to open is not correct. Moreover, although many biological processes occur at minimal force, the fact that TRR is known to act on UFBs, which are often assumed to be under tension (as the reviewer indicates below), suggests that even the application of elevated forces might be of particular physiological relevance.

This is particularly problematic for the case of BTTR where in figure 6 B it appears that the gate opening is in fact larger at low forces, closer to where the enzyme may in fact operate, rather than smaller as is observed at higher forces, which are well beyond the range of forces where the enzyme could operate.

We thank the reviewer for highlighting this point. At very low forces (<5 pN), BLM does indeed appear to increase rather than reduce the size of the open TRR-ssDNA gates. However, as we explain in the new Supplementary Note 3, the absolute change in length at these low forces is difficult to determine using our assay and thus we can only reliably interpret data above ~5 pN. We would also like to point out that the apparent lengthening of ssDNA induced by BTRR binding at forces <5 pN is similar to that observed for BLM binding alone (Supplementary Fig. 7b). Therefore, this effect seems to be predominantly caused by BLM binding to ssDNA, and not an effect of BLM altering TRR gate-opening. Moreover, as the reviewer notes in a later comment, BLM instead alters the entire energy landscape of the TRR-ssDNA gate opening process, whereas dsT-DNA simply induces a length shift. We acknowledge that this was not explained clearly in our original manuscript: while it is true that BLM effectively shortens the TRR-ssDNA open gate at elevated forces, it is more accurate to state that BLM changes the mechanical properties of the open gate, most likely by reducing the flexibility of the hinge. This represents important new structural insight into the TRR-BLM-ssDNA complex. Accordingly, we have revised our manuscript to make this point much clearer in the Results section ('*BLM alters the mechanical properties of the TRR-ssDNA gate*') as well as in the 5th paragraph of the Discussion section.

2. There is a lot of speculation about the impact of the observations on the decatenation activity of the TRR or BTTR complex, but these are not tested through biochemical experiments. Given the issues raised above, the conclusions from the single-molecule measurements are problematic, but these conclusions would nevertheless be relatively straightforward to test directly.

It is important to stress that the goal of our study is to provide fundamental insight into the mechanistic properties of TRR-ssDNA gate opening and reveal how co-factors such as magnesium, T-DNA and BLM influence these properties. Such detailed mechanistic information would be extremely difficult to obtain using bulk biochemical approaches. For example, bulk assays have been used previously to demonstrate that the presence of BLM has a stimulating effect on TRR catenation (Yang *et al.*, JBC, 2010) without being able to shed light on the precise mechanism underpinning this. Moreover, by placing our results in the context of the available literature, we believe that it is reasonable to speculate on the wider significance of our findings in the Discussion section of our manuscript. For example, we suggest that dsT-DNA may be relevant for the resolution of pre-catenanes and that the specific function of TRR / BTRR *in vivo* might be dependent on the local concentration of dsDNA versus ssDNA. We very much hope that our findings will inspire and inform future biochemical experiments that could test and explore these hypotheses further, but this would constitute an extensive research program that is beyond the scope of a single manuscript. If the reviewer and editor feel that this part of the Discussion section, which was meant to stimulate thinking about future research directions, is too speculative, we are happy to remove it.

3. The results raise some interesting and perhaps troubling questions concerning the putative roles of the TTR and BTTR complex in resolving anaphase bridges or other DNA links that are under elevated levels of stress or tension. The results suggest that the topoisomerase III gate cannot close against a force of ~5 pN or perhaps less. The authors fail to address this seeming contradiction between their

data and the physiological roles of this complex. The results in this respect are surprising and intriguing, but this aspect is largely ignored.

We agree with the reviewer that this is an intriguing point. The physiological conditions associated with UFBs are complex and are the focus of ongoing research in a number of labs. For example, the exact tension applied to UFBs *in vivo* is unknown and could conceivably vary depending on the action of proteins such as PICH that interact with UFBs. Moreover, although most TRR-ssDNA gates are open at forces from at least 5 pN, transient gate closing is still likely to occur, even at elevated tensions, and this could perhaps also be regulated by other proteins. Thus, while our findings are intriguing, they do not necessarily contradict the proposed role of TRR in UFB resolution. TRR is also involved in several other processes *in vivo*, not least double Holliday junction dissolution. In these cases, the DNA tension is expected to be low (less than a few pN) and thus transient gate closing may be more frequent. Nonetheless, we acknowledge that these points were not sufficiently addressed in our original manuscript and we now address them in the 3rd paragraph of our revised Discussion section.

4. There are a number of technical and interpretation issues that are described in detail below but which together suggest that the interpretation of the data is somewhat superficial and the findings are less clear-cut and straightforward than they are presented.

Detailed points:

Line 293: it is not entirely clear how this technique probes the steps of the catalytic cycle. One typically associates the catalytic cycle with the kinetic steps of the cycle. This assay probes the conformational change associated with gate opening and monitors a single round of catenation mediated by the presumably locked open gate. The assay therefore probes the very last step of the process and the changes in gate opening correspond to the final step of the process, just prior to gate closure, religation and release.

For the reasons explained earlier, we strongly believe that our assay provides valuable mechanistic insight into distinct steps of the catalytic mechanism of TRR. In particular, we are able to differentiate between TRR binding, ssDNA cleavage and gate opening through (a) the use or absence of magnesium and (b) the use of TRR mutants. We are also able to differentiate T-DNA binding from catenation. No kinetic measurements are required in this case, just as a crystal structure can also yield mechanistic details without the need for kinetic information. Thus, we believe that it is justified to state that our assay is able to probe different steps of the catalytic cycle.

Line 297: This technique does not measure kinetics. This is an inaccurate statement and deters from the potential impact and significance of this work.

We agree that the use of the word kinetics is not correct in this context and we have removed this in our revised manuscript.

Line 301: The measurement of the change in gate size with dsT-DNA binding is inaccurate unless individual changes in extension were measured since it is highly unlikely that every bound topoisomerase also bound a dsT-DNA segment. I would argue that the authors have demonstrated that dsT-DNA binding induces an increase in the gate opening, but the measurement represents a lower bound on the actual increase in opening given the uncertainty in the number of bound topoisomerases that also bound dsT-DNA. The increase in opening is an interesting measurement but the caveats of the extent of this opening need to be more fully discussed. This point is addressed in the supplemental information but should be included in the main text. The authors correctly state that the increase in extension is an lower bound on the increase.

The measured increase in the size of the open TRR-ssDNA gate due to dsT-DNA binding is indeed a lower estimate, and as the reviewer indicates, we previously explained this in the Supplementary Information. We agree that it would be better to state this directly in the main text and have now added this to our revised manuscript in the Results section ('Long DNA substrates can stably interact with TRR-ssDNA').

Figure 2: Based on the nominal increase in extension per bound topoisomerase at low forces, can the authors give some sense of the scale and probability of the additional opening transition above 20 pN. Is this gradual opening a kinetic effect or is it a continuous force dependent spring like extension that kicks in above 20 pN? From the curve it appears that there is a second kink in the curve around 30 pN – does the curve above ~ 30 pN map back onto the ss-DNA curve?

The reviewer raises an interesting point regarding the lengthening regime from 15 to 30 pN for TRR-ssDNA, which we now refer to as gate widening. Based on the newly provided ‘subtraction plots’ (e.g., Fig. 2c), this lengthening must correspond to a phase transition, since at forces directly below this (from 5 to 15 pN), no lengthening is observed. However, we cannot exclude that this transition might occur over a smaller force range than what we observe if the gate widening kinetics are slightly slower than the stretching speed. Nevertheless, any potential distortion due to kinetic effects cannot be very large since at forces higher than 30 pN, the lengthening is independent of force (thus the FD-curve maps back onto the ssDNA curve, with a constant shift in length).

Figure 2B – at what force was this this extension vs fluorescence intensity measured? What range of topoisomerase 3 concentration was used for these measurements? What are the error bars on the measurements of both elongation and fluorescence intensity?

We recorded fluorescence snapshots at ~10 pN to minimize Brownian fluctuations of the construct. In this particular experiment, we used a TRR concentration of 4 nM; different protein coverages were then achieved by incubating the ssDNA in TRR for varying periods of time. We have revised our Methods section (*‘Incubation conditions for single-molecule experiments’*) to state this more clearly. Regarding the errors in Fig. 2b, the uncertainty of a single measurement (for both FD-curve measurements and fluorescence intensities) is expected to be small (~1%) and we believe that the variation in the data can best be appreciated by the spread of the data points from the linear fit.

Figure 2C- what are the mean values (+/- uncertainties) of the two Gaussians?

As mentioned above, in light of the reviewers’ comments we decided to re-evaluate our step-fitting algorithm and have accordingly re-analysed the data. This revealed that, rather than having two Gaussian distributions, the step sizes fit into a single, broader distribution of 8.5 +/- 3.8 nm. This has been corrected in the revised manuscript.

Figure 3: What concentration of ds and ssDNA were used in these example measurements and what concentration of TRR was used? Without titration or other data, the change in extension associated with ssDNA or dsDNA binding are strictly lower bounds. There is a clear increase in extension for dsDNA, but unless this extension is shown to be independent of the concentration of dsDNA then the increase in extension represents an unknown number of TRR molecules that have bound dsDNA. The authors correctly note this issue in the supplemental information (Supp note 4), but this should be more clearly stated in the main text.

As originally indicated in the Supplementary Information, we used dsT- and ssT-DNA concentrations in the range of 3-10 ng/ μ l, and a TRR concentration of 20 nM. In our experiments, T-DNA binding was saturated even under the lowest T-DNA concentrations used. This is now explained in the revised Methods section (*‘Incubation conditions for single-molecule experiments’*). The reviewer is indeed correct that the measured increase in TRR-ssDNA length due to dsT-DNA binding is a lower estimate (as we indicated previously in the Supplementary Information). As suggested, we now state this in the revised Results section of the main manuscript.

Figure 3B. The dsDNA bound DNA curve appears to have three distinct regimes – there appears to be a kink in the curve around ~9 pN and another kink in the curve around 30 pN.

To address the reviewer’s point, we now provide ‘subtraction plots’ (Fig. 3d) derived from the corresponding FD-curves which show the relative change in length of TRR-ssDNA due to dsT-DNA as a function of force. Based on this, a clear gate widening transition can be observed from 15 to 30 pN. There may, perhaps, be some minor differences in the shape of the subtraction plot at high forces when comparing TRR-ssDNA with and without dsT-DNA, but the key observations are that (a) there

is a shift in length due to dsT-DNA and (b) the dsT-DNA does not inhibit the gate widening transition.

Figure 4A. Can the authors comment on the fact that the TRR curve after the 500 mM NaCl wash overlaps the ssDNA curve to a significant extent at forces up to about 10 pN? This seems to be at odds with the observation that the TRR curve shifted to the right by a fixed amount as was the case for the initial TRR curve in green, and the similar curves in Figures 2 and 3.

We thank the reviewer for highlighting this point. In light of the reviewer's comment, we identified a small error in the baseline correction for this FD-curve. After amending this, the two curves do not overlap at lower forces.

The ss and ds T-DNA binding experiments to the open gate are the most novel and unique aspects of this work. The authors have the unique ability to probe this interaction that is typically transient and impossible to capture. Can their data be interpreted in relation to the relative affinity of the enzyme cavity for the different DNA substrates? It is also possible that the affinity of the open enzyme for ssDNA or dsDNA is different – and the authors have the possibly to probe these differences. This would probe a unique and difficult to measure aspect of the process and could indirectly shed light on the strand transfer process. These measurements would be highly meaningful and informative.

We are grateful to the reviewer for this helpful feedback. To address this, we have undertaken new experiments to probe the relative affinity of ssT-DNA versus dsT-DNA for TRR-ssDNA. To this end, we incubated TRR-ssDNA in an equimolar mixture of dsT- and ssT-DNA. Since intercalators effectively stain both dsT- and ssT-DNA, they cannot be used to differentiate their respective binding to TRR-ssDNA under the conditions used above. Therefore, we used a modified staining protocol which allowed us to exclusively stain dsT-DNA with intercalators, while bound ssT-DNA could be visualized by a 2nd staining step with free TRR (shown in a new Supplementary Fig. 4d). Using this approach, substantial binding of both dsT- and ssT-DNA to the TRR-ssDNA substrate was detected. We conclude, therefore, that dsT-DNA binding has sufficient affinity for TRR-ssDNA to compete with ssT-DNA binding. In addition, we performed another new experiment, in which dsT-DNA binding to TRR-ssDNA was probed at low forces (~2 pN). We observed a similar extent of dsT-DNA as at higher forces, indicating that elevated force is not required to induce T-DNA binding. The results of these experiments are described in the revised manuscript and presented in Supplementary Fig. 4 and discussed in Supplementary Notes 7 and 8.

Figure 5F. The caption states that “ λ -ssDNA (stained with intercalator dye)” but I think that the authors meant that the circular ssT-DNA was stained.

We thank the reviewer for bringing this mistake to our attention. We indeed meant to refer to 'circular ssT-DNA', and have now corrected this in our revised manuscript.

Figure 6B. Once again the TRR curve shown in this FD curve appears to behave very differently than the ssDNA curve. There appears to be a significant kink in the curve at ~ 9-10 pN followed by a convex curvature to ~25 pN. It seems that TRR behaves differently in each of these measurements and does not consistently simply shift the ssDNA FD curve the to right as stated in Figure 2.

We acknowledge that there may be some minor variations in the FD-curves of TRR-ssDNA from molecule to molecule, but in general, they show two key features: (i) a force-independent gate open state between at least 5 pN and 15 pN and (ii) a force-dependent gate widening that occurs from 15 to 30 pN. However, we fully accept that this was not sufficiently clear from the FD-curves alone. To address this, we now show the corresponding 'subtraction plots' in our revised manuscript for the most relevant FD-curves (Figs. 2c, 3d, 5b, 6c/f). These display the length of TRR-ssDNA relative to that of bare ssDNA as a function of force, and demonstrate that the above key features are reproducibly observed for TRR in all cases other than for BTRR alone.

Furthermore, the addition of Bloom's helicase induces changes in extension that are substantially more subtle than simply decreasing the extension. Indeed, at low force BLM seems to increase the extension, and then depending on the applied force, there seems to be a variable change in the relative extension between the red and green curves. These changes in relative extension indicate that

the overall response to force has changed. This is not simply a change in the average gate opening but a complete reshaping of the force-dependent opening of the complex. The most striking observation is that at low force, the most relevant in vivo since it is closer to the force at which the gate can presumably close, the extension has increased with the addition of BLM. The complexity of the BTTR vs the TRR curve begs the question as to how the change in extension between the two complexes was determined. Is this an average difference between the two curves or an extreme value, as represented by the red arrow? Given this curve, it seems to be a misstatement to characterize the difference between TRR and BTTR as simply an decrease in gate opening extent – particularly since the opposite appears to be the case at lower forces where the enzyme could actually close the gate. It would be illustrative to see the difference in extension plotted as a function of force.

The reviewer is entirely correct that BLM alters the overall mechanical properties of TRR-ssDNA rather than inducing a simple shift in length. As discussed above, we have revised our manuscript accordingly such that we focus on this aspect and no longer on the reduction in length. The reviewer makes an excellent suggestion for plotting the difference in extension between the different constructs as a function of force. We now include these ‘subtraction plots’ in our revised version of Fig. 6c and 6f. These illustrate more clearly that, while TRR-ssDNA exhibits (i) a force-independent lengthening from 5 to 15 pN and (b) a gate widening transition from 15 to 30 pN, the change in extension due to BLM is more complex than a simple shortening. The presence of BLM alters the TRR gate flexibility substantially such that a clear distinction between gate opening and gate widening is not possible.

Along the same lines, the claim in figure 2 is that the TRR curve overlaps that of ssDNA but with an offset, if this is generally true then it would be instructive to fit the curves and report on this offset and also the degree to which the TRR and BTTR curves are indeed represented by an off-set ssDNA curve rather than something more complex.

We have now added such ‘subtraction plots’ to accompany all relevant FD-curves in the main figures. As our revised Fig. 2c demonstrates, a simple shift in the length occurs for TRR-ssDNA from at least 5 pN to 15 pN, indicating a force-independent length increase over this range, followed by a further lengthening (due to gate widening) from 15 to 30 pN. Generally, similar trends are observed when the TRR-ssDNA is bound by dsT- and ssT-DNA (see Fig. 3d), with the sole exception of BTTR-ssDNA, as highlighted above. Likewise, the use of subtraction plots allows us to demonstrate that, in contrast to TRR-ssDNA, most *Ec*Topo1-ssDNA gates are closed at forces below 10 pN (as shown in our revised version of Fig. 5b).

Supp figure 1. The abbreviations (AOTF, CMOS, EIS, etc) should be spelled out in the caption.

We have revised the caption accordingly.

Supp Figure 2. There are no kinetic measurements in this figure so the title is misleading. At best panels d and e support the claim that the opening measurements are at equilibrium, but there are no actual kinetic measurements. In panel a there is a small but reproducible downward curvature in the FD curves between ~ 10 and 24 pN, which is not consistent with a ssDNA stretching curve, this is particularly clear in panel d, but less pronounced in panel e. together these suggest that there are internal motions or transitions that are occurring as a function of force that represent a more complex process than a simple gate opening.

We agree with the reviewer that the term ‘kinetic’ is incorrect, and have thus deleted it in the revised manuscript. Regarding the variation in the TRR-ssDNA FD-curves, we believe that the reviewer is referring to the ‘shoulder’ (‘gate widening’ transition) occurring from ~15 to 30 pN. While there may be some variation from molecule to molecule (perhaps due to slightly different protein coverages), we believe that our newly provided ‘subtraction plots’ demonstrate the key features more clearly.

Supp figure 3. If I understand panel B correctly, then there is likely an excess of dsDNA on the two optically trapped beads that may interact with TRR. In this image these interactions are visualized but it seems likely or possible that this occurs in all of the experiments since the DNA substrate begins as a double stranded DNA and is mechanically manipulated to create a ssDNA. In this case, it seems that both beds could be bound to excess dsDNA molecules that could interact with TTR or BTTR or

topo I and alter the reported results. In panel e both fits are labeled “y” in principle one of them should be “x”.

The reviewer is correct that without proper precautions there is a chance that dsDNA molecules on the beads can interact with the tethered TRR-ssDNA. Nevertheless, as we now explain in the Methods section (*‘Incubation conditions for single-molecule experiments’*), we minimized this risk by ensuring that a tension of ~5 pN was applied to the tethered TRR-ssDNA construct when moving between different channels. Under these conditions, we occasionally observed crosslinking between dsDNA and TRR-ssDNA (in <10% of constructs), which resulted in a significant shortening of the construct (by >>1 μm). However, as can be appreciated from the reproducibility of the FD-curves in Supplementary Fig. 2a, such artificially shortened constructs are easily identifiable and excluded from the analysis. Regarding the labelling in panel e, we thank the reviewer for highlighting this mistake; one of the fits should indeed have been labelled as ‘x’ and we have corrected this in our revised draft.

Supp Figure 6 a. Given difference calculated, or was the difference in length averaged over different forces? Please the differences between the TRR and BTTR curves, how was the difference in length measured? At what force was the indicate how the difference in length was obtained from the FD curves that have different shapes and different distances between them, cf panel d.

As mentioned above, we agree that it is an oversimplification to state that BLM binding results only in a narrowing of the gate. In our original manuscript, we calculated the average length difference between TRR- and BTTR-ssDNA tethers at a force of 20 pN. However, since BLM does not induce a straightforward shift in length, we believe it is no longer relevant to calculate the difference in length (in agreement with the reviewer, see above) and have thus removed the original Supplementary Fig. 6a from the manuscript.

In sum, despite the overall high quality of the data and the results, I am not convinced that the work sheds light on the catalytic cycle of topoisomerase III under physiological conditions and therefore fails to contribute significantly to, or meaningfully advance, the field.

We hope that our revised manuscript, which includes a substantial number of new experiments, further data analysis and better explanations will reassure the reviewer that our study provides relevant insight into the gate mechanics of TRR-ssDNA and how co-factor binding can regulate this. A crucial improvement in our revised manuscript has been the presentation of ‘subtraction plots’ (stimulated by the reviewer’s comments), which quantify how the TRR-ssDNA length compares with that of bare ssDNA as a function of force. These demonstrate more clearly that TRR-ssDNA gate opening is independent of force (from at least 5 pN to 15 pN) and that most TRR-ssDNA gates are open by 5 pN. We believe that this provides compelling evidence that the open gate state is relevant *in vivo*. Moreover, our finding that dsT-DNA increases the gate size, while BLM alters the overall mechanical properties of TRR-ssDNA could be of direct relevance to processes such as UFB resolution and double Holliday junction dissolution.

Reviewers' Comments:

Reviewer #1:

Remarks to the Author:

The revised manuscript of Bakx et al. addresses well the previous concerns. The manuscript is improved and some important corrections have been added, for example showing that the length distribution is unimodal and not bimodal. In general all the corrections and additions are strong and improve the manuscript, but I still have two minor issues.

1. The suggested experiments with magnesium are good and help confirm what was observed before, notably in Gunn et al. They do not provide new information, as stated in one of the replies to the third reviewer, but confirmatory information. This is still important, but it should be made clear that the role of magnesium in the cleavage step was already observed before. Previous biochemical experiments failed to note the role of magnesium in cleavage, it was single molecule experiments that clarified it. The current results again show the power of single molecule experiments to observe important but difficult effects that escape bulk measurements.

2. The role of force gate opening is still unclear. The authors say that they only observed it at forces above 5 pN, which they consider small. 5 pN is not such a small force. It is large enough to overcome thermal motion, as they pointed out in the manuscript, and stretch DNA. If gate opening is stochastic, which may very well be the case, then even a low force would alter the distribution. As the authors point out in their response to reviewer 3, in the case of the *E. coli* enzymes the force applied in the gate opening experiments was much larger. Some of the concerns related to the role of force in the experiments should be addressed more directly in the discussion, pointing out the possible bias introduced by even low forces and the possibility that alternative explanations are still possible and not completely ruled out by their observations.

Reviewer #2:

Remarks to the Author:

The authors have responded helpfully to our comments. We appreciate the additional detail on the step fitting, noting that a more careful review yielded a revised result that is still consistent with the original conclusions. The authors have also done a good job characterizing the force dependence of the gate opening, yielding a plausible model for gate opening in the absence of force. The revised manuscript provides significant new insights into gate opening by this large molecular complex.

Reviewer #3:

Remarks to the Author:

In this revised manuscript, the authors provide additional data and addressed some of the points that were raised by the referees. I still have some reservations concerning the conclusions drawn from the approach with multiple bound enzymes since they do not report on the kinetics of gate opening and there are some missed opportunities to take advantage of the remarkable combination of optical trapping with fluorescence to quantify the behavior of the topoisomerase I enzymes with single- or double stranded DNA bound to the open conformation. Nonetheless, the overall findings are interesting and will potentially benefit the field. I recommend publication in *Nature Communications* after the authors address the following questions and elaborate in the discussion.

1. What was the pulling rate for the force-displacement curves that are the basis of the subtraction plots? Given the reliance on the lack of hysteresis as indicative of the lack of gate opening and closing kinetics, the pulling rate is an important parameter to include in the methods.

2. The channel designation (Supplementary Table 1) may contain an error. The second row entry for CH VI should be S rather than E, or CH V should be S rather than E.

3. In Fig. 3: Could the authors indicate either in the main text or figure caption the forces at which the T-ss/T-ds DNA capture by TRR (Fig. 3) was performed? I assume that the stretching curves were measured after TDNA capture. Also for Figure 4, can the full force trajectory be included in the caption for the catenation reactions – there is a high force loading force, then a transient decrease in force to allow the catenation reaction to occur followed by a high salt wash to remove bound proteins and DNA – describing that in the figure caption could make the process more clear.

4. In Fig. 5: The authors measured how Mg²⁺ affects the gate opening by stretching ssDNA with and without Mg²⁺ in the buffer. Based on the data shown in Fig. 2f and supp. fig. 6b, the authors concluded that cleavage by TRR and EcTopI, but not gate opening, were dependent on Mg²⁺. However, this finding (at least for EcTop1) is at odds with previous work (e.g., Domanico, P. L. & Tse-Dinh, Y. C. J. *Inorg. Biochem.* 1991, and Sissi, C., et al *Gene* 2013.) that demonstrated cleavage in the absence of Mg²⁺. Could the authors discuss these inconsistencies in more detail in the discussion?

5. In Fig. 6: Regarding the estimated TRR bound on DNA between TRR and BTRR, the authors assume comparable TRR density for two cases based on the FI measurement (Supp. Fig. 7a). However, is it possible that TRR could be localized via BLM considering TRR-BLM can physically interact which effectively reduces number of cleavage viable TRR? Do the authors have data to support the relative binding affinities of BLM and TRR to ssDNA or some other means of distinguishing between these two possibilities? If not then this caveat should be included in the description and interpretation of these results.

In conclusion, the revised manuscript is an improvement over the previous version. Although the biological import of the findings on the extent of gate opening at high forces remains tenuous, the results related to the binding and catenation of ssDNA and dsDNA to the open TRR and EcTop1 enzymes are intriguing and will likely be of interest to the community.

Response to Reviewers' comments

Reviewer #1 (Remarks to the Author):

The revised manuscript of Bakx et al. addresses well the previous concerns. The manuscript is improved and some important corrections have been added, for example showing that the length distribution is unimodal and not bimodal. In general all the corrections and additions are strong and improve the manuscript, but I still have two minor issues.

1. The suggested experiments with magnesium are good and help confirm what was observed before, notably in Gunn et al. They do not provide new information, as stated in one of the replies to the third reviewer, but confirmatory information. This is still important, but it should be made clear that the role of magnesium in the cleavage step was already observed before. Previous biochemical experiments failed to note the role of magnesium in cleavage, it was single molecule experiments that clarified it. The current results again show the power of single molecule experiments to observed important but difficult effects that escape bulk measurements.

We thank the reviewer for highlighting this. We have amended our manuscript to make this clear, and now acknowledge Gunn et al. in the Discussion (2nd paragraph).

2. The role of force gate opening is still unclear. The authors say that they only observed it at forces above 5 pN, which they consider small. 5 pN is not such a small force. It is large enough to overcome thermal motion, as they pointed out in the manuscript, and stretch DNA. If gate opening is stochastic, which may very well be the case, then even a low force would alter the distribution. As the authors point out in their response to reviewer 3, in the case of the E. coli enzymes the force applied in the gate opening experiments was much larger. Some of the concerns related to the role of force in the experiments should be addressed more directly in the discussion, pointing out the possible bias introduced by even low forces and the possibility that alternative explanations are still possible and not completely ruled out by their observations.

We agree with the reviewer that we cannot completely rule out the possibility that the size of the open gate is different in the absence of tension than it is at forces > 5 pN. We now state this clearly in the revised Results section (end of subsection 'Direct observation of TRR gate opening on ssDNA') and the revised Discussion section (end of 1st paragraph).

Reviewer #2 (Remarks to the Author):

The authors have responded helpfully to our comments. We appreciate the additional detail on the step fitting, noting that a more careful review yielded a revised result that is still consistent with the original conclusions. The authors have also done a good job characterizing the force dependence of the gate opening, yielding a plausible model for gate opening in the absence of force. The revised manuscript provides significant new insights into gate opening by this large molecular complex.

Reviewer #3 (Remarks to the Author):

In this revised manuscript, the authors provide additional data and addressed some of the points that were raised by the referees. I still have some reservations concerning the conclusions drawn from the approach with multiple bound enzymes since they do not report on the kinetics of gate opening and there are some missed opportunities to take advantage of the remarkable combination of optical trapping with fluorescence to quantify the behavior of the topo IA enzymes with single- or double stranded DNA bound to the open conformation. Nonetheless, the overall findings are interesting and will potentially benefit the field. I recommend publication in Nature Communication after the authors address the following questions and elaborate in the discussion.

1. What was the pulling rate for the force-displacement curves that are the basis of the subtraction plots? Given the reliance on the lack of hysteresis as indicative of the lack of gate opening and closing kinetics, the pulling rate is an important parameter to include in the methods.

We now state the pulling rate (2-5 $\mu\text{m/s}$) in the revised Methods section, in a new subsection titled 'Incubation procedure for single-molecule experiments'.

2. The channel designation (Supplementary Table 1) may contain an error. The second row entry for CH VI should be S rather than E, or CH V should be S rather than E.

We thank the reviewer for highlighting this error. We have rectified this in our revised SI.

3. In Fig. 3: Could the authors indicate either in the main text or figure caption the forces at which the T-ss/T-ds DNA capture by TRR (Fig. 3) was performed? I assume that the stretching curves were measured after TDNA capture. Also for Figure 4, can the full force trajectory be included in the caption for the catenation reactions – there is a high force loading force, then a transient decrease in force to allow the catenation reaction to occur followed by a high salt wash to remove bound proteins and DNA – describing that in the figure caption could make the process more clear.

We agree that this information should be stated clearly in the manuscript. Given the complexity of the experimental protocols, we feel that describing this information in the captions would be too technical and distracting for the general reader. Instead, we have compiled a new Methods subsection titled 'Incubation procedure for single-molecule experiments' which describes the incubation protocols and force application procedures in detail. Additionally, we now state in the relevant sections of the Results that information regarding incubation protocols is described in the Methods. We have also revised the caption of Fig. 4 to state that we reduced the DNA tension in the high salt buffer in order to help protein unbinding.

4. In Fig. 5: The authors measured how Mg^{2+} affects the gate opening by stretching ssDNA with and without Mg^{2+} in the buffer. Based on the data shown in Fig. 2f and supp. fig. 6b, the authors concluded that cleavage by TRR and EcTopI, but not gate opening, were dependent on Mg^{2+} . However, this finding (at least for EcTop1) is at odds with previous work (e.g., Domanico, P. L. & Tse-Dinh, Y. C. J. Inorg. Biochem. 1991, and Sissi, C., et al Gene 2013.) that demonstrated cleavage in the absence of Mg^{2+} . Could the authors discuss these inconsistencies in more detail in the discussion?

We agree that this discrepancy should be mentioned in the manuscript. We note that Gunn *et al.* (NSMB, 2017) also reported the requirement of magnesium for ssDNA cleavage by EcTopoI/III using single-molecule approaches. We now state this in our revised Discussion, and additionally acknowledge that the discrepancy between single-molecule and bulk studies will require future research efforts to fully resolve (2nd paragraph).

5. In Fig. 6: Regarding the estimated TRR bound on DNA between TRR and BTRR, the authors assume comparable TRR density for two cases based on the FI measurement (Supp. Fig. 7a). However, is it possible that TRR could be localized via BLM considering TRR-BLM can physically interact which effectively reduces number of cleavage viable TRR? Do the authors have data to support the relative binding affinities of BLM and TRR to ssDNA or some other means of distinguishing between these two possibilities? If not then this caveat should be included in the description and interpretation of these results.

We thank the reviewer for raising this point. We now provide additional control measurements in the new Supplementary Fig. 7b which show that we observe the same effect of BLM, independent of whether we incubate the ssDNA in TRR and BLM together or whether TRR-ssDNA is incubated in BLM separately. We believe that this provides strong evidence that BLM does not hinder the interaction of TRR with ssDNA. We now state this clearly in the revised Results section (subsection 'BLM alters the mechanical properties of the TRR-ssDNA gate').

In conclusion, the revised manuscript is an improvement over the previous version. Although the biological import of the findings on the extent of gate opening at high forces remains tenuous, the results related to the binding and catenation of ssDNA and dsDNA to the open TRR and ECTop1 enzymes are intriguing and will likely be of interest to the community.